# Transgenic NADH dehydrogenase restores oxygen regulation of breathing in mitochondrial complex I-deficient mice

Blanca Jiménez-Gómez [1,2,3,5], Patricia Ortega-Sáenz [1,2,3,5], Lin Gao [1,2,3], Patricia González-Rodríguez[1,2,3], Paula García-Flores [1,2,3], Navdeep Chandel [4] & José López-Barneo [1,2,3] ✉

The hypoxic ventilatory response (HVR) is a life-saving reflex, triggered by the activation of chemoreceptor glomus cells in the carotid body (CB) connected with the brainstem respiratory center. The molecular mechanisms underlying glomus cell acute oxygen ($O_2$) sensing are unclear. Genetic disruption of mitochondrial complex I (MCI) selectively abolishes the HVR and glomus cell responsiveness to hypoxia. However, it is unknown what functions of MCI (metabolic, proton transport, or signaling) are essential for $O_2$ sensing. Here we show that transgenic mitochondrial expression of NDI1, a single-molecule yeast NADH/quinone oxidoreductase that does not directly contribute to proton pumping, fully recovers the HVR and glomus cell sensitivity to hypoxia in MCI-deficient mice. Therefore, maintenance of mitochondrial NADH dehydrogenase activity and the electron transport chain are absolutely necessary for $O_2$-dependent regulation of breathing. NDI1 expression also rescues other systemic defects caused by MCI deficiency. These data explain the role of MCI in acute $O_2$ sensing by arterial chemoreceptors and demonstrate the optimal recovery of complex organismal functions by gene therapy.

$O_2$ is needed by eukaryotic cells to generate ATP. Adaptive cardiorespiratory responses to a lack of $O_2$ (hypoxia), such as hyperventilation and increased heart output, are thus essential for survival. These reflex responses to hypoxia, that take place over a time course of seconds, are mediated by the carotid body (CB), the prototypical acute $O_2$-sensing organ in mammals[1,2]. The CB contains neurosecretory glomus cells that express $O_2$-regulated $K^+$ channels, the inhibition of which by hypoxia leads to transmitter release to activate sensory fibers impinging on brainstem respiratory and autonomic centers[3–6]. The molecular mechanism underlying acute $O_2$ sensing by CB glomus cells has been difficult to elucidate; however, recent data have shown that mutations in mitochondrial complex (MC) I and MCIV impair glomus cell responsiveness to hypoxia and the hypoxic ventilatory response (HVR)[7–9]. In particular, ablation of the gene coding NDUFS2, a core subunit required for MCI assembly and function, results in selective abolition of $O_2$ regulation of breathing mediated by dopaminergic CB glomus cells[7]. *Ndufs2*-null mice also show dwarfism, early death and motor and metabolic disorders reflecting affectation of other catecholaminergic territories[7,10]. Mutations in the *Ndufs2* gene produce a Leigh-like syndrome in humans[11]. A fundamental issue to elucidating the molecular determinants of arterial chemoreception is to find out which of the MCI activities (NADH dehydrogenase, electromotive force generation, or signaling) is key for acute $O_2$ sensing. In addition, it is also unclear whether a complete reconstitution of MCI, which contains 45 subunits in mammals[12,13], is required for the restoration of $O_2$-dependent breathing control. We have addressed these questions here

[1]Instituto de Biomedicina de Sevilla (IBiS), Hospital Universitario Virgen del Rocío/CSIC/Universidad de Sevilla, Seville, Spain. [2]Departamento de Fisiología Médica y Biofísica, Facultad de Medicina, Universidad de Sevilla, Seville, Spain. [3]Centro de Investigación Biomédica en Red sobre Enfermedades Neurodegenerativas (CIBERNED), Madrid, Spain. [4]Department of Pediatrics, Northwestern University, Chicago, IL, USA. [5]These authors contributed equally: Blanca Jiménez-Gómez, Patricia Ortega-Sáenz. ✉e-mail: lbarneo@us.es

by studying conditional *Ndufs2* knockout (MCI-deficient) mice expressing the single-molecule yeast NADH dehydrogenase NDI1. This enzyme, like MCI, catalyzes NADH oxidation and ubiquinone (CoQ) reduction, but it does not pump protons that contribute to ATP production[14]. NDI1 has been expressed in mammalian cells to rescue MCI enzymatic activity[15], and it was reported recently that expression of NDI1 prolongs lifespan in a mouse model of Leigh syndrome although the ataxic motor symptoms and seizures were not improved[16]. Here, we show that NDI1 expression in MCI-deficient glomus cells in vivo completely restores the $O_2$-dependent regulation of breathing, as well as glomus cell and mitochondrial responses to hypoxia. Dwarfism, lethality, and systemic metabolic alterations were also corrected by transgenic NDI1 expression.

## Results and discussion

### NDI1 expression restores the hypoxic ventilatory response and prevents systemic pathologies in embryonic MCI-deficient mice

Mice carrying *Ndufs2 floxed* allele were used to generate conditional *Ndufs2* knockout mice (*Ndufs2* flox/-;Cre, KO mice) in which the Cre recombinase was controlled by the tyrosine hydroxylase (TH) promoter[7]. In addition, mice carrying lox-STOP-lox sites preceding yeast *Ndi1* and green fluorescent protein (GFP) sequences targeted to the *Rosa26* locus[16] were also utilized to generate KO/NDI1 mice expressing NDI1 in NDUFS2-deficient dopaminergic CB glomus cells (Fig. 1a). Mice with mitochondrial mutations were subjected to whole body plethysmography to test for changes in the $O_2$-dependent regulation of breathing. In comparison with wild type (WT) mice, a complete abolition of the HVR was seen in KO mice and this response was almost completely recovered in KO/NDI1 mice (Fig. 1b, c; Supplementary Fig. 1a, b). Basal breathing frequency under normoxic conditions was consistently reduced (-10%) in KO mice, compared to WT, and recovered in KO/NDI1 mice (Fig. 1b). This phenomenon, also observed in other mouse models with disruption of mitochondrial ETC in CB glomus cells[17], could be a consequence of a decreased respiratory drive impinging on brain centers in mice with $O_2$-insensitive CBs. It could be also a reflection of general alterations in metabolism or in temperature control of KO mice (see Supplementary Fig. 2). However, the ventilatory response to hypercapnia was preserved in KO or KO/NDI1 mice (Fig. 1d, e; Supplementary Fig. 1b), indicating that the respiratory network and its modulation by central and peripheral $CO_2$ chemoreceptors were not significantly affected by mitochondrial mutations in TH-expressing cells. These data showed that selective impairment of the $O_2$-dependent regulation of breathing induced by disrupted MCI function in arterial chemoreceptor cells is reversed by the transgenic expression of the single-molecule yeast NADH dehydrogenase.

Conditional NDUFS2-deficient (KO) mice appeared normal in general appearance, weight, and survival rate during the first 3–4 weeks after birth, as reported before[7,10]. Absence of phenotype in this early postnatal period was not surprising as MCI proteins commonly have lifetimes of 20–40 days[18]. Thereafter, KO mice showed several systemic alterations, unrelated to CB acute $O_2$ sensing, due to damage to hypothalamic neurons and other MCI-deficient cell types in which TH was expressed transiently during development[7,10,16,19,20]. The transgenic expression of NDI1 decreased hyperlactatemia (Fig. 1f) and prevented reduced lifespan (Fig. 1g) and dwarfism (Fig. 1h, i; Supplementary Fig. 2a–c) in MCI-deficient (KO) mice. Other systemic alterations observed in MCI-deficient mice, such as decrease in body temperature or exaggerated hematocrit increase under exposure to chronic hypoxia, were also corrected by NDI1 expression (Supplementary Fig. 2d, e).

### Metabolic reprograming induced by MCI deficiency in chemoreceptor cells is reversed by NDI1 expression

We demonstrated by immunohistochemistry (Fig. 2a) and qPCR (Fig. 2b) the selective expression of NDI1 in CB cells of KO/NDI1 mice,

as well as the abolition of NDUFS2 protein and mRNA expression in glomus cells of KO and KO/NDI1 mice (Fig. 2c, d). Immunocytochemical evidence of NDI1 (GFP) expression and down-regulation of *Ndufs2* mRNA and protein expression were also confirmed in catecholaminergic superior cervical ganglion (SCG) of KO and KO/NDI1 mice (Supplementary Fig. 3a, b). A decrease in MCI activity in NDUFS2-deficient cells independently of NDI1 expression was also confirmed in SCG (Supplementary Fig. 3c). It is known that in NDUFS2-deficient glomus cells basal NADH levels are higher than in WT mice, suggesting an alteration in oxidative phosphorylation (OXPHOS), although ATP levels, measured in whole CB tissue, are not decreased[7]. To determine whether the mitochondrial deficit in OXPHOS triggered adaptive metabolic changes supporting the survival of glomus cells, we measured the cytosolic ATP/ADP ratio by two-photon laser scanning microscopy using the genetically-encoded sensor PercevalHR[10,21]. This study revealed that in contrast to glomus cells from WT mice, in which inhibition of MCV by oligomycin caused a sudden drop in ATP levels, in glomus cells from KO mice the ATP/ADP ratio fell only in response to inhibition of glycolysis with 2-deoxiglucose (2-DG) (Fig. 2e, f). Indeed, in MCI-deficient (KO) cells inhibition of MCV caused a small, but measurable, increase in ATP levels, indicating that in these cells, a constant glycolytic production of ATP is used by MCV functioning in reverse mode to maintain the proton electrochemical gradient across the mitochondrial inner membrane. Remarkably, OXPHOS was restored in glomus cells from KO/NDI1 mice (Fig. 2e, f), suggesting that once NADH dehydrogenase activity is recovered by NDI1 expression, proton pumping by MCIII and MCIV compensates the lack of electromotive force generation by MCI.

### NDI1 expression rescues responsiveness of MCI-deficient chemoreceptor cells to hypoxia

The responsiveness of single chemoreceptor cells to hypoxia was studied by monitoring quantal catecholamine release (Fig. 3a). Both hypercapnia ($CO_2$) and high $K^+$ elicited secretory activity in glomus cells from WT and KO mice, however the response to hypoxia was selectively inhibited in cells from KO mice (Fig. 3b, c, i–l). In contrast, glomus cell responsiveness to hypoxia was recovered in KO/NDI1 mice (Fig. 3d, j), with a full secretory response reversibly blocked by extracellular $Cd^{2+}$ (Fig. 3e), suggesting that as in WT cells, this effect depended on $Ca^{2+}$ influx[5,22]. Recovery of sensitivity to hypoxia in KO/NDI cells occurred in a broad range of $O_2$ tensions (Supplementary Fig. 4). At low $PO_2$ values, there was a trend for KO/NDI1 cells to show a rate of hypoxia-induced dopamine secretion higher than that in WT cells (Fig. 3j). The cytosolic $Ca^{2+}$ signal induced by hypoxia in glomus cells, abolished in KO mice[7], was also rescued in NDI1-expressing cells (Supplementary Fig. 5). In WT cells, rotenone, a selective MCI blocker that does not affect NDI1[15,23], increased glomus cell secretory activity similarly to that induced by hypoxia alone (Fig. 3f). However, rotenone occluded any effect of hypoxia when the two conditions were applied simultaneously[7,22] (Fig. 3f, m, n). While hypoxia and rotenone did not have any effect on MCI-null cells (KO mice) (Fig. 3g, m, n), the hypoxic activation of KO cells expressing NDI1 was nevertheless maintained in the presence of rotenone (Fig. 3h, m, n). These results indicate that MCI-dependent acute $O_2$ sensing, which is abolished in glomus cells of KO mice, was completely restored by the expression of rotenone-insensitive NDI1. WT/NDI1 mice, which express NDI1 in the CB and have normal levels of NDUFS2 (Supplementary Fig. 6a–c), showed normal systemic ventilatory responses to hypoxia and hypercapnia (Supplementary Fig. 7a–d). Moreover, activation by hypoxia, hypercapnia, and high $K^+$ of single glomus cells from the CBs of these animals was also qualitatively normal (Supplementary Fig. 8a–f). However, similar to KO/NDI1 cells, there was a trend for responsiveness to hypoxia in WT/NDI1 to be higher than in WT cells (Supplementary Fig. 8d), while WT/NDI1 cells were

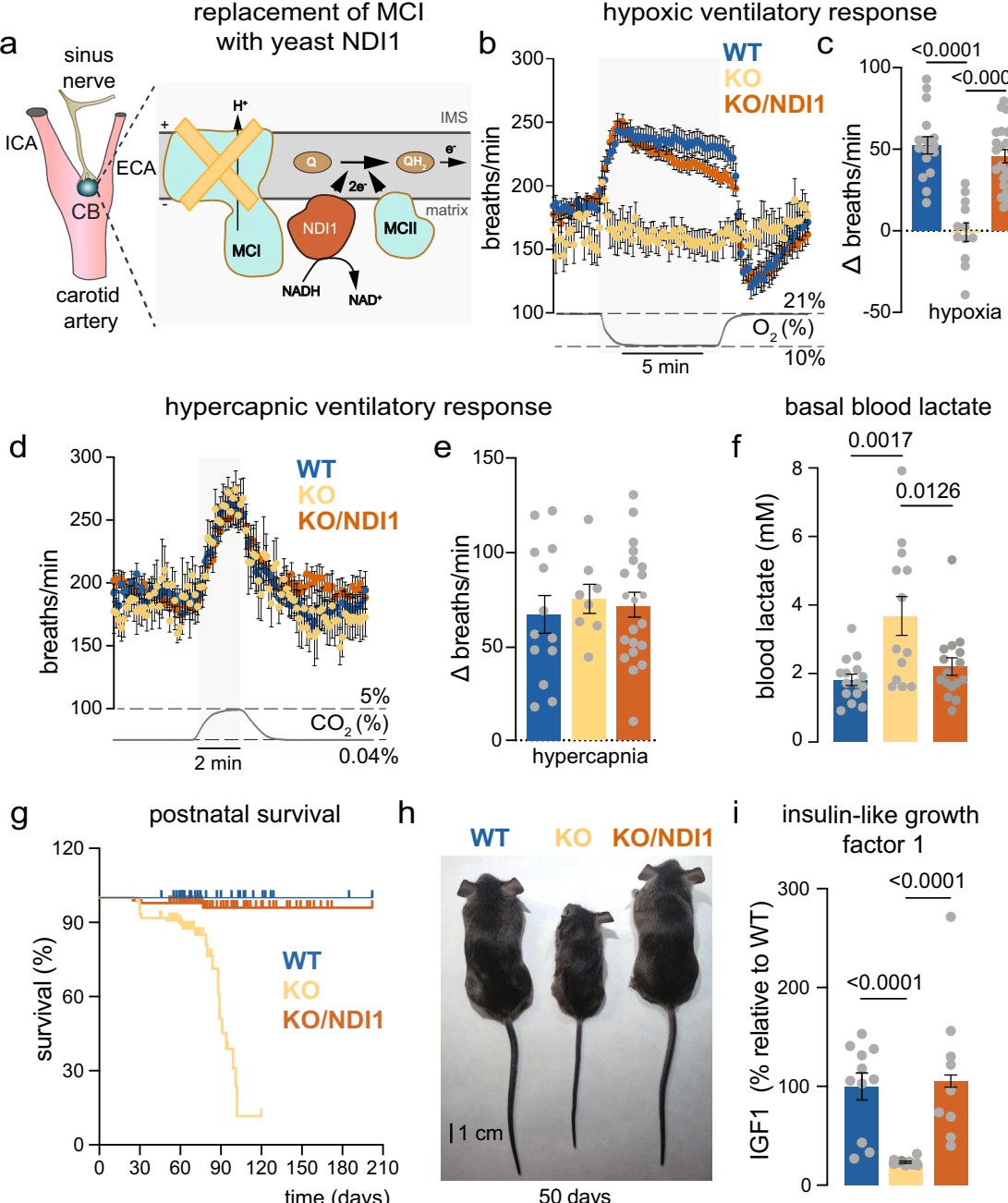

**Fig. 1 | Rescue of the hypoxic ventilatory response and systemic defects by transgenic NDI1 in mitochondrial complex I-deficient mice. a** Left. Scheme illustrating the carotid artery bifurcation and location of the carotid body (CB). ICA internal carotid artery, ECA external carotid artery. Modified from ref. [20]. Right. Replacement of mitochondrial complex I (MCI) with NDI1 in the KO/NDI1 mouse. IMS: Intermembrane space; Q and QH₂: oxidized and reduced forms of ubiquinone, respectively. **b** Hypoxic (10% O₂) ventilatory response (HVR, breaths/min) of the mouse strains studied (WT, $n = 16$; KO, $n = 12$; KO/NDI1, $n = 22$). Each data point is represented as mean ± SEM. Changes in % O₂ with time are represented at the bottom. **c** Average increase in respiratory frequency induced during hypoxia (10% O₂, Δ breaths/min) in the three mouse models studied (WT, $52 \pm 5$, $n = 16$; KO, $-1 \pm 6$, $n = 12$; KO/NDI1, $46 \pm 4$, $n = 22$). **d** Average hypercapnic ventilatory response (5% CO₂, breaths/min) of the mouse strains studied (WT, $n = 13$; KO, $n = 8$; KO/NDI1, $n = 21$).

Each data point is represented as mean ± SEM. Changes in % CO₂ with time are represented at the bottom. **e** Increase in respiratory frequency induced by hypercapnia (5% CO₂, Δ breaths /min) in the three mouse models studied (WT, $67 \pm 10$, $n = 13$; KO, $75 \pm 8$, $n = 8$; KO/NDI1, $71 \pm 7$, $n = 21$). **f** Basal lactate plasma levels, in mM, from WT (blue, $1.8 \pm 0.16$, $n = 15$), KO (yellow, $3.7 \pm 0.6$, $n = 13$) and KO/NDI1 (brown, $2.2 \pm 0.25$, $n = 16$) mice. **g** Kaplan-Meier postnatal survival plots of WT (blue, $n = 36$), KO (yellow, $n = 118$) and KO/NDI1 (brown, $n = 102$) mice. **h** Representative images of WT, KO and KO/NDI1 female mice at ~50 postnatal days of age. **i** Serum levels of insulin-like growth factor-1 (IGF-1) measured in the mice strains studied normalized to WT. WT, $100 \pm 14$, $n = 11$; KO, $23 \pm 1$, $n = 10$; KO/NDI1, $105 \pm 20$, $n = 11$. Data are presented as mean ± SEM with statistically significant *P* values (>0.05) superimposed. In (**c**, **e**, **f**, **i**), *P* values were calculated by one-way ANOVA followed by Tukey's multiple comparisons post hoc test. Source data are provided as a Source Data file.

insensitive to rotenone (Supplementary Fig. 8b, g, h). Therefore, it seems that NDI1 cannot only replace MCI but when the two enzymes are present NDI1 overrides MCI and becomes the predominant mitochondrial NADH/CoQ oxidoreductase in glomus cells.

**Mitochondrial signaling of hypoxia in MCI-deficient and NDI1-expressing chemoreceptor cells**

It has been suggested that hypoxia causes a slowdown of the mitochondrial electron transport chain (ETC) in glomus cells,

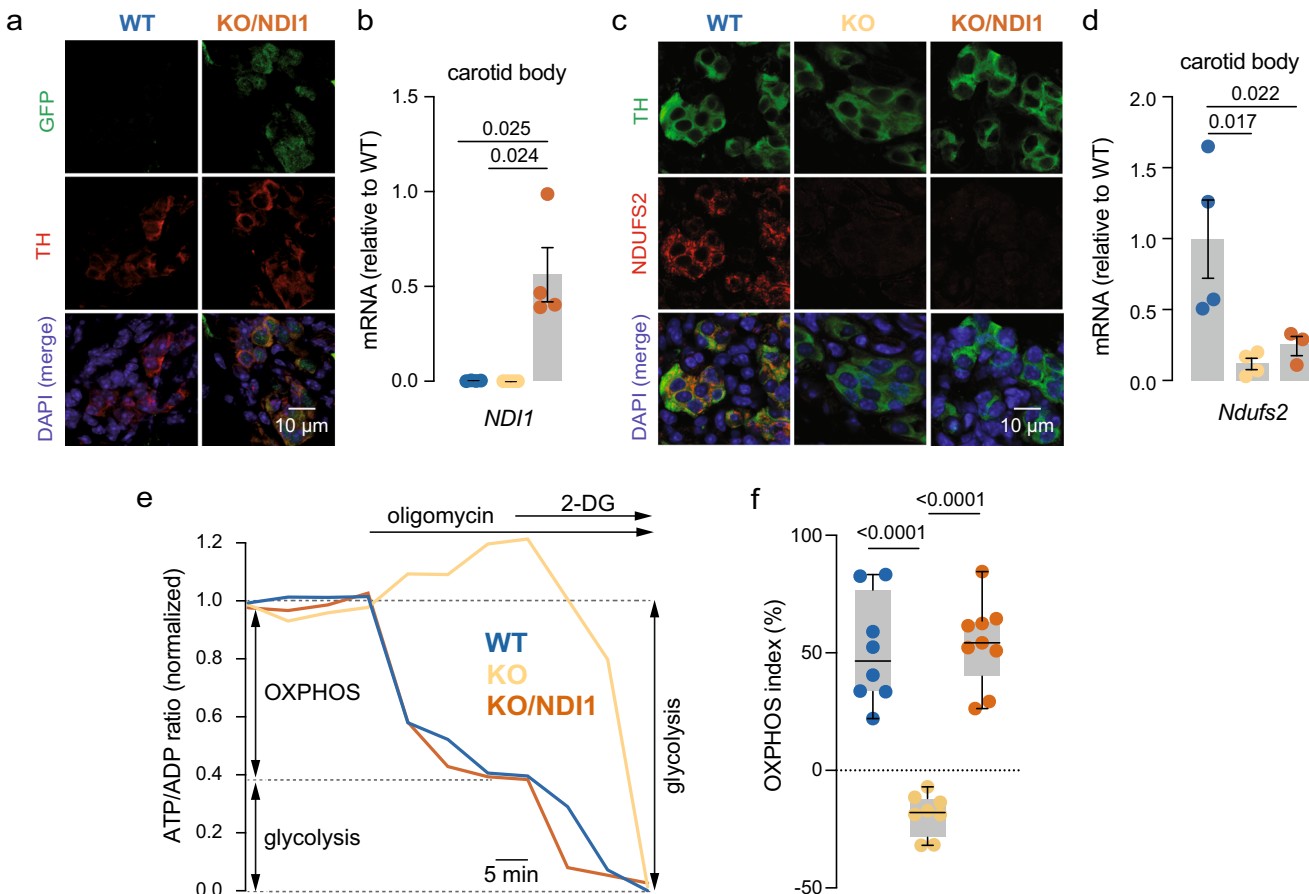

**Fig. 2 | NDI1 expression and mitochondrial bioenergetic restoration in complex I-deficient carotid body glomus cells. a** Histological sections of the carotid body (CB) from WT (left) and KO/NDI1 (right) mice illustrating colocalization of NDI1 expression (green fluorescent protein, GFP) and tyrosine hydroxylase (TH). DAPI was used to stain nuclei. Similar immunocytochemical studies were performed in $n = 4$ mice for each genotype. Calibration bar (10 μM) applies to all panels. **b** *NDI1* mRNA levels, relative to WT, in CB samples from WT mice (blue dots, $0.030 \pm 0.0009$, $n = 4$ replicates/group), KO mice (yellow dots, $0 \pm 0$, $n = 4$ replicates/group) and KO/NDI1 mice (brown dots, $0.56 \pm 0.14$, $n = 4$ replicates/group). Data are expressed as mean ± SEM. P values calculated by one-way ANOVA followed by Newman-Keuls multiple comparisons test are indicated. **c** Immunostaining of CB sections illustrating NDUFS2 protein expression in TH positive cells from WT mice (left) and the disappearance in CB cells from KO (middle) and KO/NDI1 (right) mice. Similar immunocytochemical studies were performed in $n = 4$ mice for each genotype. Calibration bar (10 μM) applies to all panels. **d** *Ndufs2* mRNA levels, relative to WT, in CB samples from WT mice (blue dots, $1 \pm 0.3$, $n = 4$ replicates/group), KO

mice (yellow dots, $0.1 \pm 0.04$, $n = 4$ replicates/group) and KO/NDI1 mice (brown dots, $0.24 \pm 0.07$, $n = 3$ replicates/group). Data are expressed as mean ± SEM. *P* values calculated by one-way ANOVA followed by Newman-Keuls multiple comparisons test are indicated. **e** Representative time-lapse measurements of the ATP/ADP ratio estimated in glomus cells in CB slices expressing PercevalHR obtained from the various mouse models studied. Oligomycin (10 μM) and 2-deoxyglucose (5 mM glucose in the external solution was replaced with 5 mM 2-DG) were applied to determine the contribution of oxidative phosphorylation (OXPHOS) and glycolysis to the ATP/ADP ratio. Recordings are normalized to the value at the onset of each experiment. **f** Distribution of the OXPHOS index (OXPHOS/OXPHOS + glycolysis) in glomus cells from the mouse models studied. WT (blue, $n = 8/5$ cells/mice), KO (yellow, $n = 8/4$ cells/mice) and KO/NDI1 (brown, $n = 9/4$ cells/mice). The boxplots represent median (middle line), 25th, 75th percentile (box), and largest and smallest values range (whiskers). Statistically significant *P* values, calculated by non-parametric Kruskal-Wallis tests followed by Dunn's post hoc test. Source data are provided as a Source Data file.

which leads to an increased $CoQH_2/CoQ$ ratio, slowdown/reversal of MCI, and rapid increases in NADH and ROS levels[7,9] (Fig. 4a). As these mitochondrial signals modulate membrane ion channel activity, their abolition in MCI-deficient glomus cells results in the inhibition of responsiveness to hypoxia[7]. We studied whether MCI-deficient cells expressing NDI1, a single-molecule NADH/CoQ oxidoreductase with only one or two steps in internal electron transfer and not coupled to proton pumping[23–25], can also generate hypoxic mitochondrial signals (Fig. 4b). NADH levels were reversibly increased by hypoxia in WT glomus cells, a response that was practically abolished in KO glomus cells as shown previously[7] (Fig. 4c, d, j). NDI1 expression restored basal NADH levels, which were increased in NDUFS2-deficient cells[7], to near normal (Fig. 4i) as well as hypoxia-activated NADH signals, which had a larger amplitude than NADH signals recorded in WT cells (Fig. 4e, j). Hypoxia-induced NADH signals in KO/NDI1 cells were also faster

(1.3 arbitrary units (au)/s, $n = 17$) than those in WT cells (0.5 au/s, $n = 32$, $p < 0.0001$). In WT cells, rotenone increased NADH and occluded any further effect of hypoxia (Fig. 4f, k, l)[8], however this agent did not have any effect on KO (Fig. 4g, k, l) or KO/NDI1 (Fig. 4h, k, l) cells. Moreover, the responsiveness of KO/NDI1 cells to hypoxia was unaffected by rotenone (Fig. 4h, l). These data indicate that replacement of MCI with NDI1 not only supports mitochondrial ETC with a rotenone-insensitive NADH dehydrogenase activity, but that during hypoxia NDI1 is also slowed down/reversed at a higher rate than that of MCI, thereby leading to larger and faster NADH signals. In support of this concept, hypoxic NADH signals in glomus cells from WT/NDI1 mice were also of larger amplitude than those of WT cells, and insensitive to rotenone (Supplementary Fig. 9a-f). These findings further suggest that NADH dehydrogenase activity is mainly mediated by NDI1 when MCI and NDI1 are expressed together in glomus cells.

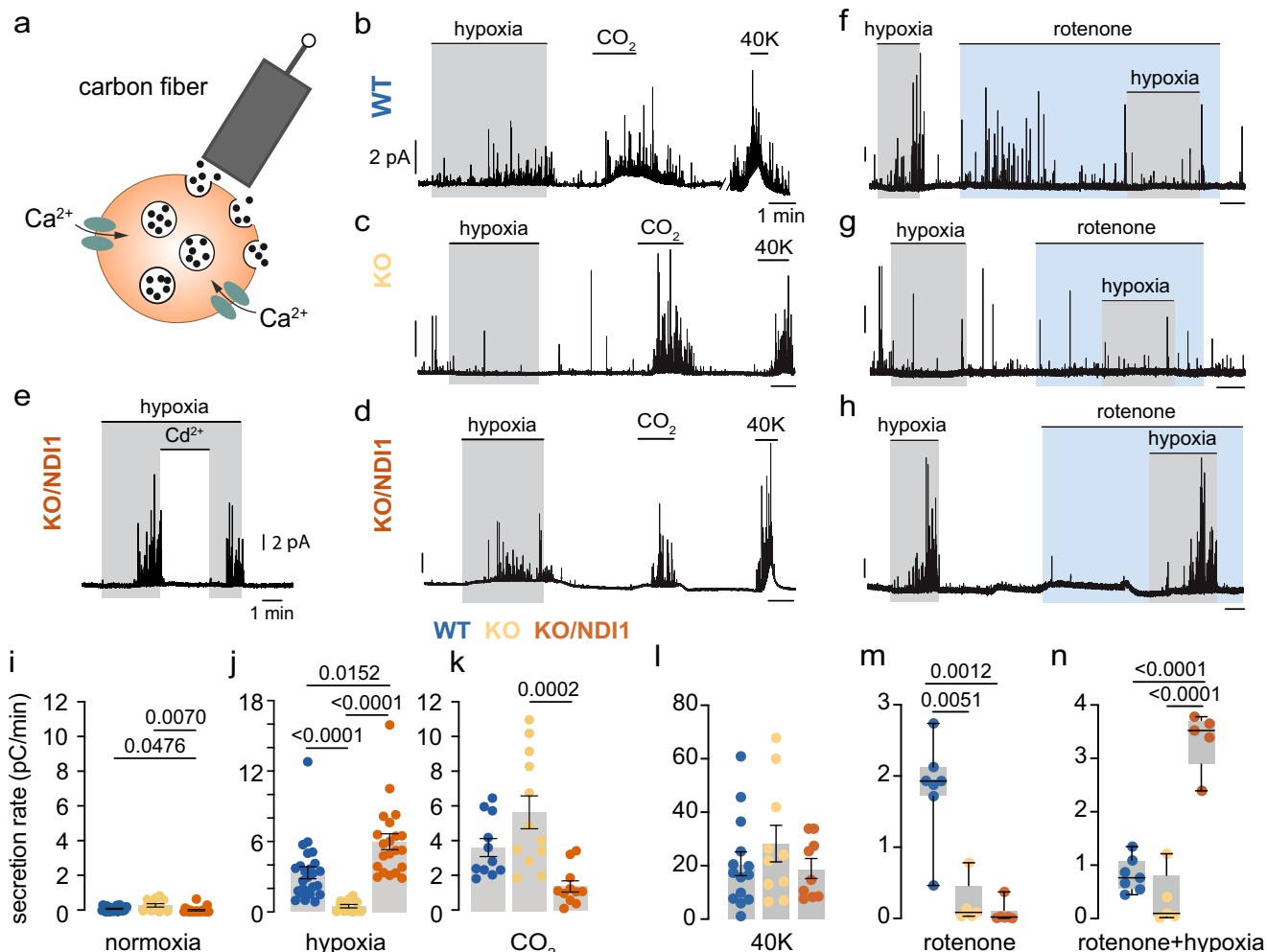

**Fig. 3 | Rescue of the secretory response to hypoxia by transgenic NDI1 in mitochondrial complex I-deficient carotid body glomus cells. a** Scheme illustrating the amperometric recording of quantal dopamine release from a single glomus cell. Modified from ref. [20]. **b–d** Secretory responses of glomus cells in carotid body (CB) slices induced by hypoxia ($O_2$ tension ~15 mmHg), hypercapnia (20% $CO_2$) and depolarization with potassium (40 mM K) in wild type (**b**), KO (**c**) and KO/NDI1 (**d**) mice. The values of the calibration bars in (**b**), also apply to (**c**, **d**). **e** Inhibition of the secretory response induced by hypoxia in cells from KO/NDI1 mice by blockade of $Ca^{2+}$ channels with cadmium (200 μM). **f–h** Secretory responses of glomus cells in CB slices to hypoxia and rotenone (5 μM) in WT (**f**), KO (**g**), and KO/NDI1 (**h**) mice. Calibration bars as indicated in (**b**). **i–k** Quantification (in picoCoulombs/min) of the basal secretion rate in normoxia and the secretory response to hypoxia and hypercapnia ($CO_2$) of the mouse models studied. Normoxia (**i**): WT, $0.19 \pm 0.03$, $n = 26/21$ cells/mice; KO, $0.37 \pm 0.1$, $n = 12/7$ cells/mice; KO/NDI, $0.10 \pm 0.04$, $n = 21/10$ cells/mice. Hypoxia (**j**): WT, $3.3 \pm 0.5$, $n = 25/21$ cells/mice; KO, $0.5 \pm 0.1$, $n = 12/7$ cells/mice; KO/NDI1, $5.9 \pm 0.7$, $n = 21/10$ cells/mice. $CO_2$, (**k**): WT, $3.6 \pm 0.5$, $n = 11/9$ cells/mice; KO, $5.6 \pm 1$, $n = 12/7$ cells/mice; KO/NDI1,

$1.4 \pm 0.3$, $n = 11/6$ cells/mice. Data are expressed as mean ± SEM with individual values superimposed. Statistically significant *P* values, calculated by one-way ANOVA followed by Tukey's multiple comparisons post hoc test, are represented in each panel. **l** Average secretion rate (pC/min) induced by 40 mM KCl in glomus cells from WT mice ($20.8 \pm 4.5$, $n = 14/7$ cells/mice); KO mice ($28.3 \pm 6.8$, $n = 10/7$ cells/mice); KO/NDI1 mice ($19.0 \pm 3.7$, $n = 9/6$ cells/mice). Data are expressed as mean ± SEM. **m**, **n** Boxplots representing the distribution of secretion rates induced by rotenone (**m**) and hypoxia in the presence of rotenone (**n**) in glomus cells in CB slices from the mouse models studied. WT mice (rotenone: $n = 7/7$ cells/mice; rotenone + hypoxia: $n = 7/7$ cells/mice); KO mice (rotenone: $n = 5/3$ cells/mice; rotenone + hypoxia: $n = 5/3$ cells/mice) and KO/NDI1 mice (rotenone: $n = 6/4$ cells/mice; rotenone + hypoxia: $n = 6/4$ cells/mice). The boxplots indicate median (middle line), 25th, 75th percentile (box), and largest and smallest values range (whiskers). In (**m**), *P* values were calculated with two-tailed Mann–Whitney test. In (**n**), two-tailed unpaired t tests were used. Source data are provided as a Source Data file.

We next used microfluorimetry with genetically-encoded probes[26] to study whether the mitochondrial intermembrane space (IMS) ROS signal, characteristic of WT glomus cells exposed to hypoxia[8,9,17] (see Fig. 4a, b), could also be recorded in cells expressing NDI1. The IMS ROS signal, which was abolished or strongly reduced in amplitude in KO cells (Fig. 5a, b, g), was also rescued in NDUFS2-deficient and NDI1-expressing glomus cells (Fig. 5c, g). In contrast with the IMS signal, ROS in the mitochondrial matrix, which are produced to a significant extent by the activity of tricarboxylic acid (TCA) cycle dehydrogenases[27,28], decreased in WT and KO cells in response to hypoxia (Fig. 5d, e, h), probably reflecting the decrease in $O_2$ availability[8,9] (Fig. 5i). However, a

robust hypoxia-induced reversible increase in matrix ROS was recorded in KO cells expressing NDI1 (Fig. 5f, h). This observation suggests that, similar to MCI, ROS are generated when NDI1 is slowed down/reversed during hypoxia secondarily to a sudden increase in $QH_2$. However, ROS produced by NDI1, a peripheral protein anchored to the matrix side of the inner mitochondrial membrane[23,24], are directed to the matrix from where they seem to diffuse to the IMS (Fig. 5j). In agreement with this idea, WT/NDI1 glomus cells, in which NDI1 is the predominant mitochondrial NADH dehydrogenase, also responded to hypoxia with an increase in IMS and matrix ROS (Supplementary Fig. 9g–j).

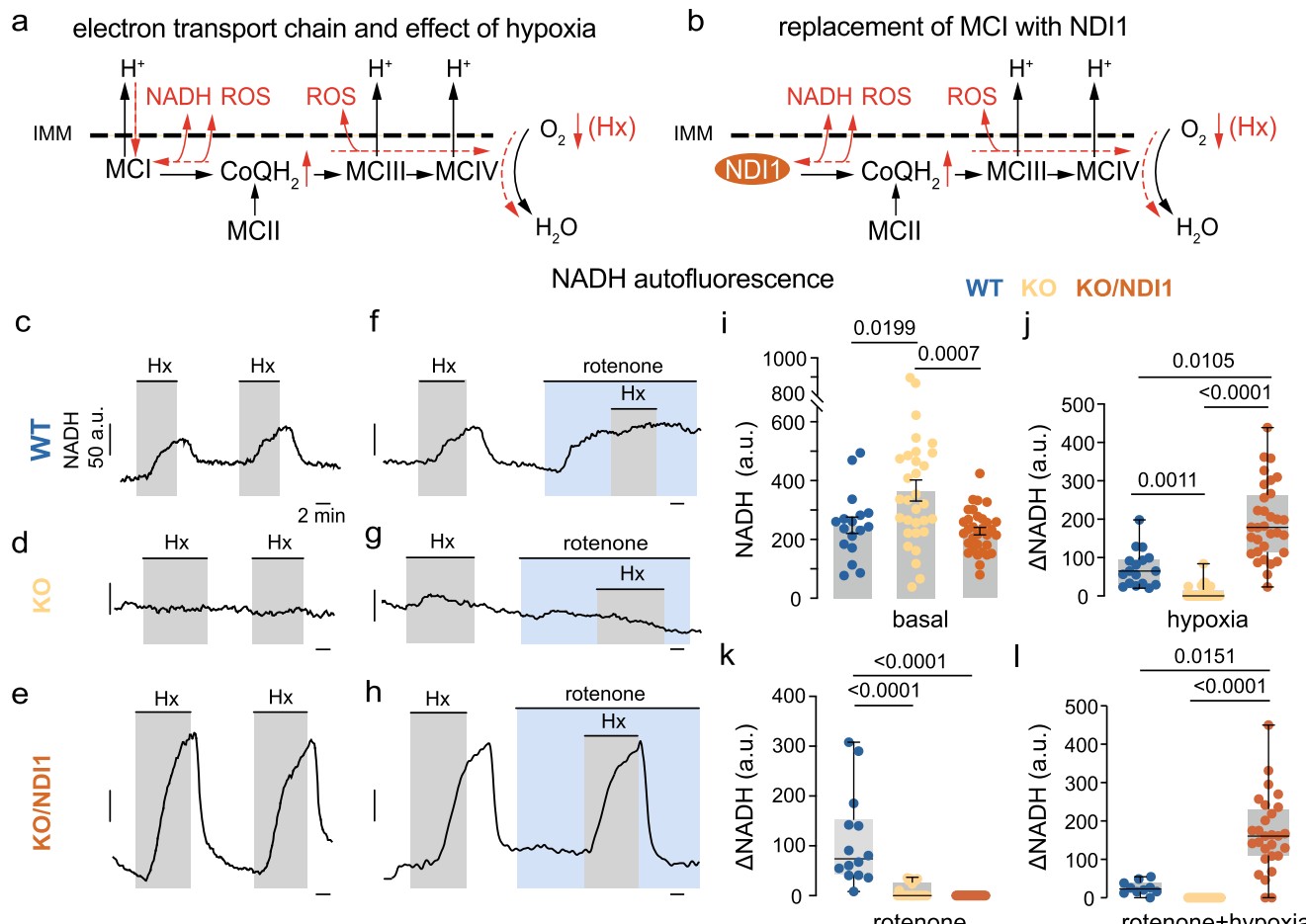

**Fig. 4 | Mitochondrial NADH and hypoxia signaling in complex I-deficient and NDI1-expressing glomus cells. a** Scheme representing the electron transport chain (ETC) in glomus cells and its dynamic changes in response to hypoxia (Hx, red lines and symbols). IMM, inner mitochondrial membrane. **b** Scheme representing the ETC of glomus cells in which MCI is replaced with NDI1. Dynamic changes in hypoxia are represented by red lines and symbols. **c–e**, Representative recordings of changes in NADH autofluorescence recorded in response to hypoxia (Hx, PO$_2$ ~15 mmHg) in dispersed glomus cells from WT (**c**), KO (**d**) and KO/NDI1 (**e**) mice. Calibration bars as indicated in (**c**). **f–h** Microfluorimetric recordings illustrating changes in NADH induced by hypoxia (Hx), rotenone (1 μM) and the application of hypoxia in the presence of rotenone in dispersed glomus cells from WT (**f**) KO (**g**) and KO/NDI1 (**h**) mice. Calibration bars as indicated in (**c**). **i** Average basal levels of NADH (in arbitrary units, a.u) recorded in glomus cells from WT mice (254 ± 28,

$n = 17/6$ cells/mice), KO mice (372 ± 36, $n = 31/6$ cells/mice) and KO/NDI1 mice (234 ± 13, $n = 32/6$ cells/mice). Data are expressed as mean ± SEM. *P* values calculated by one-way ANOVA followed by Tukey's multiple comparisons post hoc test. **j–l** Boxplots representing increase in NADH autofluorescence in response to hypoxia (**j**, $n$ = panel **i**), rotenone (**k**, WT, $n = 14/6$; KO, $n = 25/6$, KO/NDI1, $n = 28/6$ cells/mice) and rotenone plus hypoxia (**l**, WT, $n = 10/6$; KO, $n = 18/6$, KO/NDI1, $n = 28/6$ cells/mice) recorded in dispersed glomus cells from WT (blue dots), KO (yellow dots) and KO/NDI1 (brown dots) mice. The boxplots in (**j**, **k**, **l**) indicate median (middle line), 25th, 75th percentile (box), and largest and smallest values range (whiskers). Indicated *P* values were calculated with Kruskal-Wallis test followed by Dunn's multiple comparisons post hoc test. Source data are provided as a Source Data file.

## NDI1 restores acute O$_2$-sensing and survival in MCI-deficient adult mice

To test the ability of transgenic NDI1 expression to prevent disruption of O$_2$ regulation of breathing induced in adulthood, we generated mice that ubiquitously expressed tamoxifen (TMX)-inducible Cre recombinase as well as the floxed *Ndufs2* allele and/or *Ndi1* transgene. In this manner, we were able to study CB acute O$_2$ sensing of WT littermates (ESR-WT mice) in comparison with NDUFS2-deficient mice with (ESR-KO/NDI1) and without (ESR-KO) the simultaneous expression of NDI1 (Supplementary Fig. 10). Early death after TMX treatment in ESR-KO mice, due to a generalized loss of MCI function[7,8], was not observed in the ESR-KO/NDI1 mice, which survived for several months until they were used for experiments (Fig. 6a). Moreover, ESR-KO/NDI1 mice also showed almost complete recovery of the HVR, which, as shown previously[8], was selectively abolished in ESR-KO cells (Fig. 6b–i). As it occurred in KO mice (Fig. 1b), basal breathing frequency under normoxic conditions was consistently reduced (-15%) in ESR-KO mice

compared to ESR-WT mice, and was recovered in ESR-KO/NDI1 mice (Fig. 6b–i). Similar to CB cells from KO/NDI1 mice (see Fig. 3), glomus cells from ESR-KO/NDI1 mice, which did not express NDUFS2 protein (Fig. 7a, b), showed a fully rescued secretory response to hypoxia (Fig. 7c–e, i–k). This response was unaffected by rotenone (Fig. 7f–h, l, m). NADH signaling in response to hypoxia was also recovered in glomus cells from ESR-KO/NDI1 mice (Supplementary Fig. 11).

The primary outcome of this paper is that the abolition of O$_2$-dependent regulation of breathing resulting from disruption of MCI in mouse arterial chemoreceptor cells is completely rescued by the transgenic expression of yeast NDI1. This remarkable functional recovery occurs at the mitochondrial, cellular, and systemic levels. NDI1 not only restores the ETC and oxidative ATP synthesis in glomus cells, but from the standpoint of mitochondrial responsiveness to hypoxia it seems to work even better than MCI. These findings have important implications for our understanding of the molecular mechanisms underlying acute O$_2$ sensing, as they demonstrate the

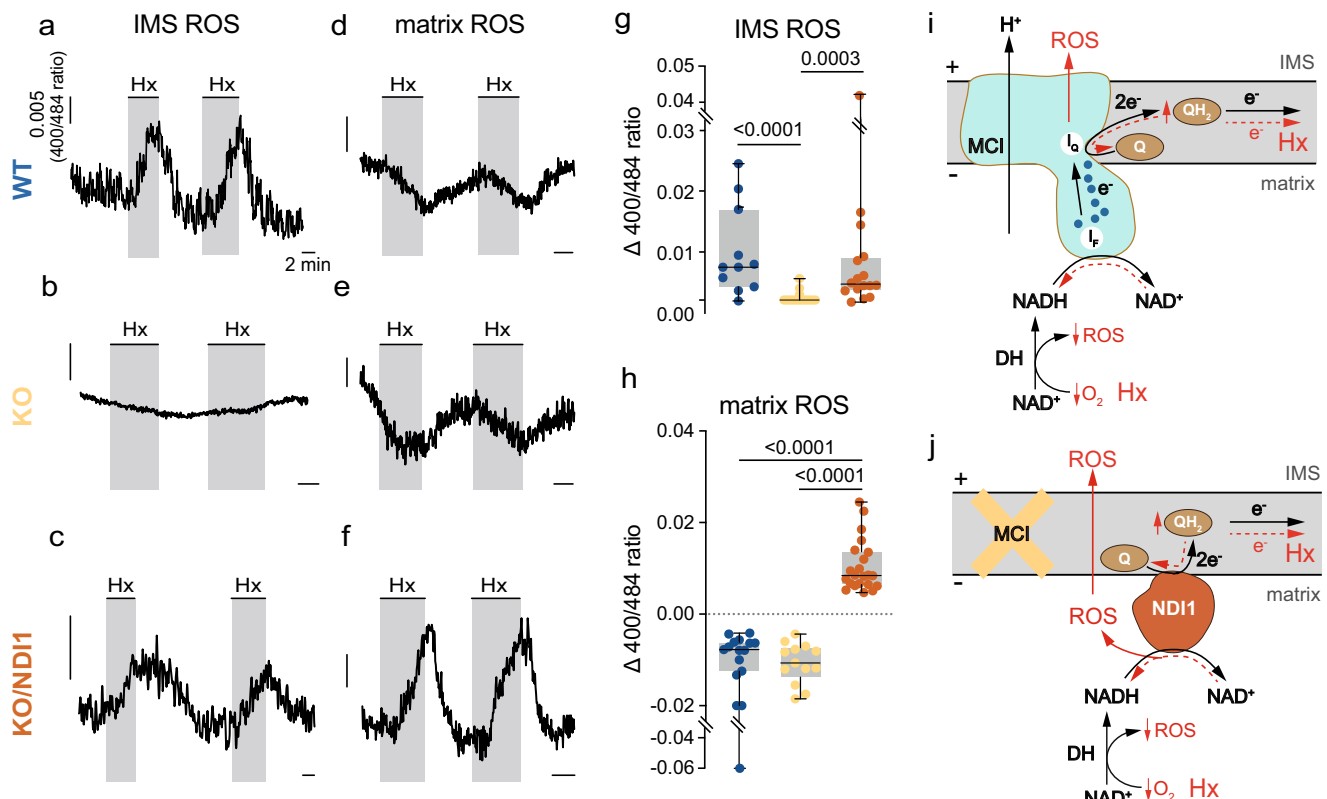

**Fig. 5 | Mitochondrial ROS compartmentalization and hypoxia signaling in complex I-deficient and NDI1-expressing glomus cells. a–f** Representative microfluorimetric recordings of changes in reactive oxygen species (ROS) at the mitochondrial intermembrane space (IMS ROS, **a**, **b**, **c**) and at the mitochondrial matrix (matrix ROS, **d**, **e**, **f**) induced by hypoxia (Hx, $O_2$ tension ~15 mmHg) in glomus cells in CB slices from WT (**a**, **d**), KO (**b**, **e**) and KO/NDI1 (**c**, **f**) mice. Calibration bars as indicated in (**a**). **g**, **h** Boxplots representing hypoxia-induced increases (increase in 400/484 ratio) in IMS ROS (**g**) and matrix ROS (**h**) in glomus cells from WT mice (blue dots, IMS ROS, $n = 11/6$ cells/mice; matrix ROS, $n = 15/8$ cells/mice), KO mice (yellow dots, IMS ROS, $n = 15/3$ cells/mice; matrix ROS, $n = 13/4$ cells/mice), KO/NDI1 mice (brown dots, IMS ROS, $n = 16/7$ cells/mice; matrix ROS, $n = 22/10$ cells/mice). All individual data points are superimposed. The boxplots indicate median (middle line), 25th, 75th percentile (box), and largest and smallest values range (whiskers). Indicated *P* values were calculated with Kruskal-Wallis test

followed by Dunn's multiple comparisons post hoc test. **i** Scheme of compartmentalized mitochondrial ROS production during hypoxia in normal glomus cells (red lines and symbols). The increase in reduced ubiquinone ($QH_2$) causes slow-down/reversal of MCI and production of ROS directed to the IMS. This IMS signal is abolished, or strongly reduced in MCI-deficient glomus cells (**b**, **g**). Matrix ROS, mainly due to the activity of dehydrogenases (DH), diminishes during hypoxia due to the decreased availability of $O_2$. This matrix ROS signal is little affected by MCI deficiency (**d**, **e**, **h**). **j** Scheme of compartmentalized mitochondrial ROS production during hypoxia (red lines and symbols) in glomus cells in which MCI is replaced with NDI1. The increase in reduced ubiquinone ($QH_2$) during hypoxia causes slowdown/reversal of NDI1 and a large burst of ROS at the matrix which predominates over the decrease in ROS produced by the dehydrogenases (DH). Matrix ROS, possibly in the form of $H_2O_2$, diffuses to the IMS. Source data are provided as a Source Data file.

essential role of MCI in the generation of hypoxia-signaling molecules (NADH and ROS), independently of its contribution to proton pumping and ATP synthesis. Our findings also indicate that a structural shift of MCI to a de-active state[29–31] with $Na^+/H^+$ antiporter activity or reduction of cysteine residues in NDUFS2[32], suggested to occur during acute hypoxia in other cell types, are not required for hypoxia signaling in WT glomus cells. These results strongly suggest that acute $O_2$ sensing is an integral function of the glomus cell ETC in which the slowdown/reversion of MCI and generation of mitochondrial signals (NADH and ROS) is combined with an accelerated provision of substrates by the TCA cycle and the low apparent affinity of MCIV for $O_2$[33].

The current study also provides unprecedented data on how the ETC reacts to hypoxia in chemoreceptor cells expressing NDI1 in comparison with WT or MCI-deficient (KO) cells. An unexpected observation was that during exposure to hypoxia NDI1 generates NADH signals that are larger and faster than those generated by MCI. This suggests that the rate of NADH dehydrogenase activity of NDI1 is much higher than that of MCI. The NADH/CoQ oxidoreductase activity of MCI, which requires movements of CoQ/CoQH₂ in and out the quinone cage coupled with structural changes to drive proton pumping, is a relatively slow, rate limiting, step in the mitochondrial

ETC[13,34]. In contrast, NDI1 has a much simpler catalytic cycle with only one or two intermediate steps and close proximity between the NADH and CoQ binding sites[24]. Another surprising finding related to the enzymatic role of NDI1 is that the presence of this yeast enzyme practically abolishes MCI function, even in WT glomus cells. A possible explanation for this phenomenon is that by capturing most of the NADH available, NDI1 drives MCI to a de-active state[13,35,36]. This fact could also explain an intriguing effect of NDI1 expression as there is a trend for glomus cells from KO/NDI1 and WT/NDI1 mice to be less sensitive to $CO_2$. Glomus cell activation by $CO_2$ depends on intracellular acidification by carbonic anhydrase[1], which could possibly be inhibited by a relative cytosolic alkalization resulting from the $Na^+/H^+$ antiporter activity of de-active MCI[37]. In contrast with glomus cells from WT or KO mice, hypoxia elicited a robust increase in matrix ROS in cells expressing NDI1 (either KO/NDI1 or WT/NDI1 cells), which suggest that when NDI1 activity is slowed down during hypoxia, the enzyme is capable of producing a fast and reversible ROS signal. This is a rather unexpected observation given that a common belief is that ROS production in mitochondria is decreased by NDI1 expression[38]. However, it must be recalled that other matrix dehydrogenases (e.g., α-ketoglutarate dehydrogenase) are able to generate large amounts of

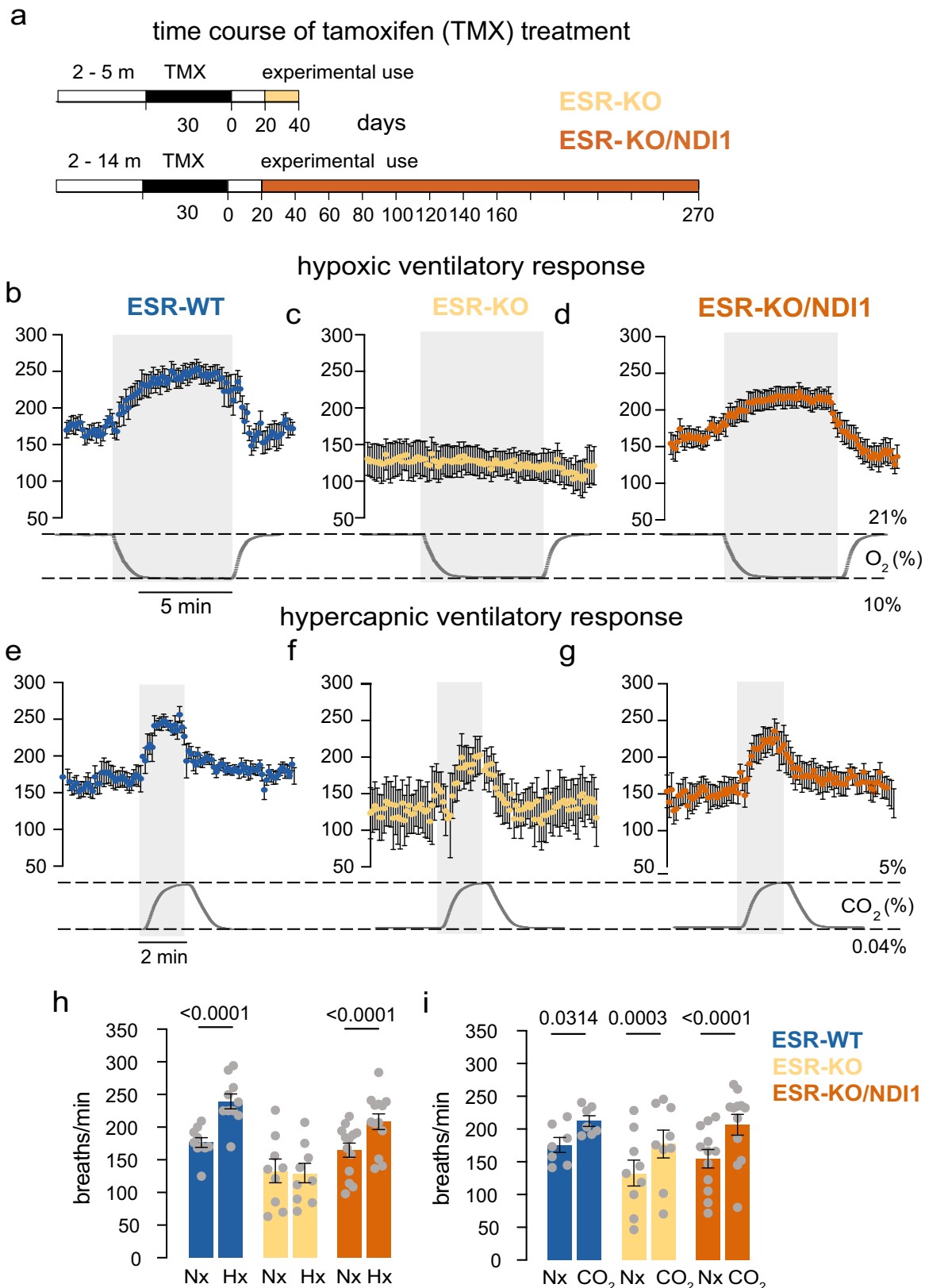

ROS[27,28]. In addition, structural studies have suggested the existence of single internal electron transfer from NADH to CoQ in NDI1 with the formation of semiquinone, which could favor ROS production when the reaction of the enzyme is slowed down/reversed[24,39]. Therefore, it seems that in glomus cell mitochondria NDI1 expression leads to de-activation of MCI. Upon exposure to hypoxia, the sudden increase in

$CoQH_2$ induces slowdown/reversion of the NDI1 reaction and the production of rotenone-insensitive NADH and matrix ROS. An additional relevant observation from our work is that transgenic NDI1, activated during either embryonic or adult life, can compensate perfectly for MCI deficiencies in vivo that affect non-dividing fully differentiated cells, such as CB glomus cells or central neurons. This raises

**Fig. 6 | Rescue of the hypoxic ventilatory response in mitochondrial complex I-deficient mice by conditional transgenic NDI1 expression in adulthood.**
**a** Scheme illustrating the time course of tamoxifen (TMX) treatment to generate the adult ESR-KO (top) and the ESR-KO/NDI1 (bottom) mice. The age at which mice were used for experiments is indicated. Note that whereas all ESR-KO mice died before 50 days after termination of tamoxifen treatment, lifespan was clearly increase in ESR-KO/NDI1 mice (**b**–**d**), Hypoxic (10% $O_2$) ventilatory response (HVR, breaths/min) of the mouse strains studied (ESR-WT, $n = 10$; ESR-KO, $n = 9$; ESR-KO/NDI1, n = 13). Changes in $O_2$ tension (%) with time are represented at the bottom. Each data point is represented as mean ± SEM. Time calibration bar in (**b**) (5 min) also applies to (**c**, **d**). **e**–**g** Hypercapnic ventilatory response (5% $CO_2$, breaths/min) of the mouse strains studied (ESR-WT, $n = 7$; ESR-KO, $n = 9$; ESR-KO/NDI1, $n = 12$). Changes in % $CO_2$ with time are represented at the bottom. Each data point is represented as mean ± SEM. Time calibration bar in (**e**) (2 min) also applies to (**f**, **g**).
**h** Average respiratory frequency (breaths /min) during normoxia (Nx, 21% $O_2$) and during hypoxia (Hx, 10% $O_2$) in the three mouse models studied. ESR-WT mice (Nx: 177 ± 7; Hx: 239 ± 11), ESR-KO mice (Nx: 131 ± 18; Hx: 127 ± 15), ESR-KO/NDI1 mice (Nx: 165 ± 11; Hx: 208 ± 12). Data are presented as mean ± SEM. Number of mice as in (**b**–**d**). Statistically significant $P$ values calculated by two-tailed paired t test are indicated. **i** Average respiratory frequency (breaths/min) recorded in basal conditions (normoxia, Nx, 0.04% $CO_2$) and during hypercapnia (5% $CO_2$) in the three mouse models studied. ESR-WT mice (Nx: 176 ± 11; $CO_2$: 212 ± 8), ESR-KO mice (Nx: 133 ± 20; $CO_2$: 177 ± 20), ESR-KO/NDI1 mice (Nx: 155 ± 14; $CO_2$: 207 ± 16). Data are presented as mean ± SEM. Number of mice as in (**e**–**g**). Statistically significant $P$ values calculated by two-tailed paired t test are indicated. Source data are provided as a Source Data file.

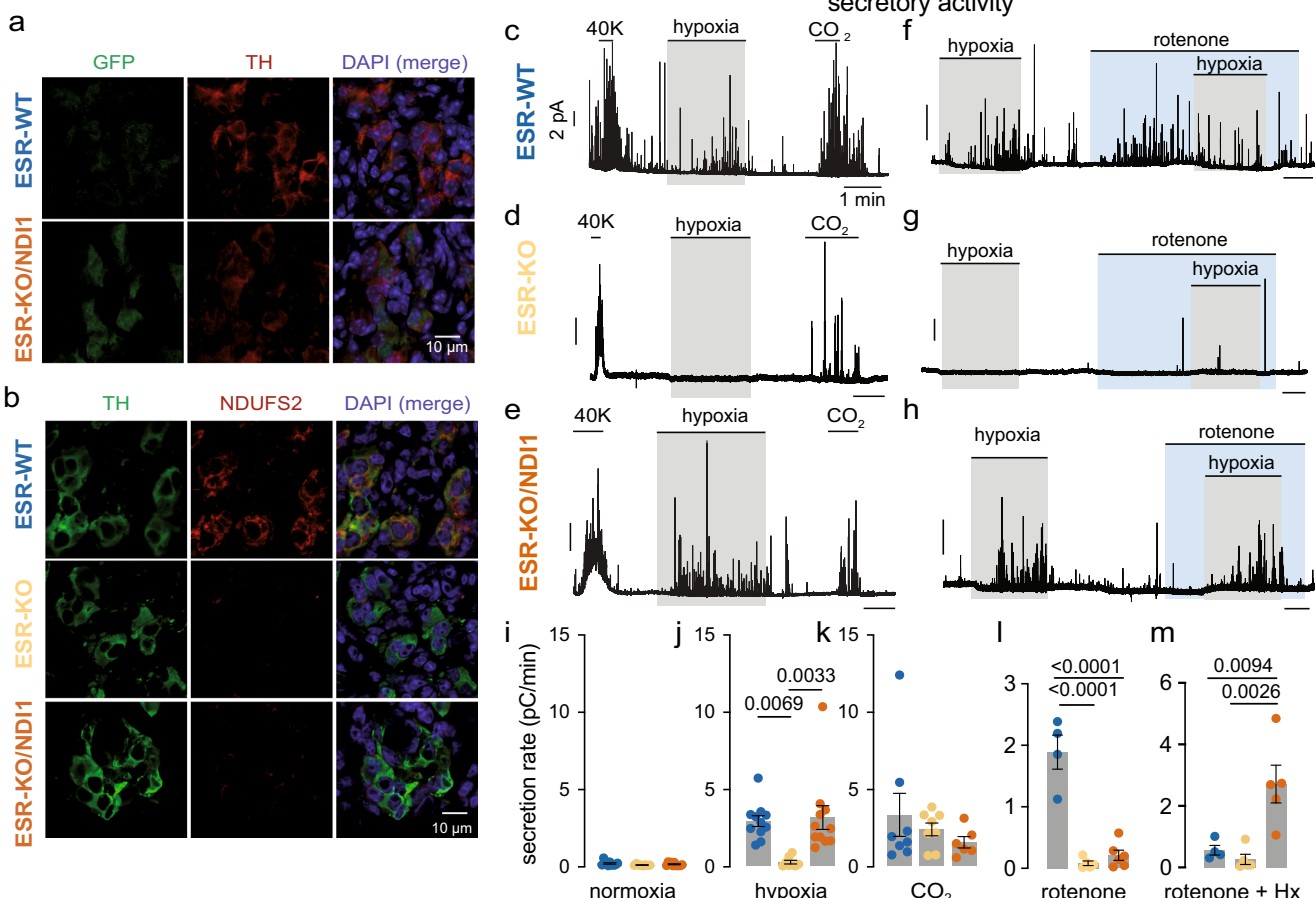

**Fig. 7 | Rescue of the secretory response to hypoxia of mitochondrial complex I-deficient glomus cells by conditional transgenic NDI1 expression in adulthood.** **a** Histological sections of the carotid body (CB) from ESR-WT (top) and ESR-KO/NDI1 (bottom) mice illustrating immunoreactivity for GFP (green, indicating NDI1 expression) and TH (red). Similar studies were performed in 4 mice for each genotype. DAPI was used to stain nuclei (blue). Calibration bar (10 µM) applies to all panels. **b** Histological sections illustrating TH (green) and NDUFS2 (red) protein expression in CB glomus cells from ESR-WT (top), ESR-KO (middle) and ESR-KO/NDI1 (bottom) mice. Nuclei were stained with DAPI (blue). Similar immunocyto-chemical studies were performed in $n = 4$ mice for each genotype. Calibration bar (10 µM) applies to all panels. **c**–**e** Representative amperometric recordings of the secretory activity induced by hypoxia ($O_2$ tension -15% mmHg) hypercapnia (20% $CO_2$) and depolarization with potassium (40 mM K) in carotid body glomus cells from ESR-WT (**c**), ESR-KO (**d**) and ESR-KO/NDI1 (**e**) mice. Calibration bars as indicated in (**c**). **f**–**h** Representative recordings of the secretory activity induced by rotenone (5 µM) and hypoxia in the presence of rotenone in glomus cells from ESR-WT (**f**), ESR-KO (**g**) and ESR-KO/NDI1 (**h**) mice. Calibration bars are indicated in (**c**). **i**–**k** Basal secretion rate (picoCoulombs/min) in normoxia and the secretory

response to hypoxia and hypercapnia ($CO_2$) of glomus cells in CB slices from the mouse models studied. In all cases $n$ = cells/mice. Normoxia (**i**): ESR-WT, 0.17 ± 0.04, $n = 11/9$; ESR-KO, 0.06 ± 0.02, $n = 8/4$; ESR-KO/NDI, 0.11 ± 0.03, $n = 11/5$. Hypoxia (**j**): ESR-WT, 2.90 ± 0.35, $n = 11/9$; ESR-KO, 0.25 ± 0.11, $n = 8/4$; ESR-KO/NDI, 3.14 ± 0.77, $n = 11/5$. $CO_2$, (**k**): ESR-WT, 3.31 ± 1.40, $n = 8/7$; ESR-KO, 2.37 ± 0.42, $n = 8/4$; ESR-KO/NDI, 1.54 ± 0.37, $n = 6/4$. Data are expressed as mean ± SEM with all data values superimposed. Statistically significant $P$ values, calculated by one-way ANOVA followed by Tukey's multiple comparisons post hoc test, are represented in each panel. **l**, **m** Average secretion rate (picoCoulombs/min) induced by rotenone (**l**) and hypoxia in the presence of rotenone (**m**) in glomus cells in CB slices from mouse models studied. In all cases $n$ = cells/mice. ESR-WT (rotenone: 1.89 ± 0.28, $n = 4/3$; rotenone + hypoxia: 0.56 ± 0.16, $n = 4/3$); ESR-KO (rotenone: 0.08 ± 0.04, $n = 5/2$; rotenone + hypoxia: 0.26 ± 0.17 $n = 5/2$) and ESR-KO/NDI1 (rotenone: 0.21 ± 0.08, $n = 6/4$; rotenone + hypoxia: 2.71 ± 0.61 $n = 5/3$). In (**l**, **m**) $P$ values represented in each panel were calculated by one way ANOVA followed by Tukey's multiple comparisons post hoc test. Source data are provided as a Source Data file.

the possibility that NDI1, or similar enzymes, could be successfully used for gene therapy in mitochondrial pathologies such as Leigh syndrome or Parkinson's disease[40,41].

Overall, our study strongly supports the view that MCI signaling in specialized mitochondria is essential for acute $O_2$ sensing in CB chemoreceptor cell. Moreover, our results demonstrate that transgenic NDI1 expression in embryonic or adult life can rescue disruption of the $O_2$-dependent regulation of breathing along with other complex organismal dysfunctions caused by MCI deficiency. The ability of a single-molecule enzyme that does not contribute directly to mitochondrial proton pumping and energy storage to substitute the activity of a 45-subunit MCI without any apparent major side-effects should stimulate the development of gene replacement therapies for human mitochondrial diseases.

## Methods

### Animal models and maintenance
Mice carrying *Ndufs2* floxed alleles (flox/flox mice)[7] were bred with those carrying CRE recombinase under the control of tyrosine hydroxylase (TH) promoter[19] to generate NDUFS2 embryonic conditional knockout mice (Ndufs2 flox/-;Cre, named as KO mice in the text and figures). In addition, mice carrying a lox-STOP-lox site preceding yeast Ndi1 and green fluorescent protein (GFP) sequences targeted to the Rosa26 locus[16] were also utilized to generate KO/NDI1 mice expressing NDI1 in NDUFS2-deficient dopaminergic CB glomus cells. Mice carrying yeast NADH dehydrogenase (NDI1) and green fluorescent protein (GFP) sequences were also bred with those carrying CRE recombinase under the control of TH promoter to express NDI1 in TH+ cells (named as WT/NDI1). Similarly, NDUFS2 adult conditional knockout mice (named as ESR-KO), and adult conditional knockout mice with NDI1 expression (ESR-KO/NDI1) were also generated, using mice carrying tamoxifen-induced CRE recombinase[42]. For microfluorimetric studies (see below) we generated mice that did not express GFP to avoid interferences. To this end, mice carrying NDI1 and GFP were bred with those carrying FLP recombinase[43] to eliminate GFP expression, which was flanked by two Frt sites. The genotype of each mouse was determined by polymerase chain reaction (PCR)[7,16].

Mice were housed at a regulated temperature ($22 \pm 1\,°C$) in a 12-hr light/12-h dark cycle with ad libitum access to food (Teklad global 14% protein, Envigo) and drink. Both male and female mice were used in the current study. Mice were either in C57BL/6 or a mixed genetic background (129 Sv:C57BL/6). To induce CRE mediated recombination, adult wildtype and conditional knockout mice ≥2 months old were fed with a tamoxifen-containing diet (TAM400/CreER, 400 mg/kg tamoxifen citrate, Envigo) for a month followed by normal diet until they were used for experiments. In animals treated with chronic hypoxia (10% $O_2$ atmosphere for 10 days), the experiments were performed using a hermetic chamber with control of $O_2$, $CO_2$ humidity and temperature (Coy Laboratory Products, Inc.)[44]. For in vitro studies, mice were sacrificed with intraperitoneal administration of a lethal dose of sodium thiopental (120 to 150 mg/kg) before tissue dissection. Dissected tissues were used for either functional, biochemical, or immunohistochemical analyses. For RNA isolation and measurement of mitochondrial complex I activity or serum levels of insulin-like growth factor, tissues were flash-frozen in liquid $N_2$ and stored at −80 °C until use. For generation of Kaplan-Meier survival plots, death events were counted when animals die spontaneously. Data points were censored when moribund and non-terminal animals were euthanized for experimentation. All procedures were approved by the Institutional Committee for animal care and use at the University of Seville (PN2019 07/04/2020/051). Handling of the animals was conducted in accordance with the European Community Council directive of 22 September 2010 (Directive 2010/63/EU) and the implementations of 5 June 2019 (Regulation 2019/2010) for the Care and Use of Laboratory Animals. Mice used in the in vivo measurements were subsequently sacrificed for the in vitro experiments in order to minimize the number of animals used in the study.

### Plethysmography
Awake unrestricted mice were placed inside plethysmography chambers (EMKA Technologies) to study respiratory function[45]. Data acquisition was performed using the IOX2 software (EMKA Technologies; RRID:SCR_022973). Chambers were filled with either air (21% $O_2$, normoxia) or a gas mixture: 10% $O_2$ (hypoxia, maintained for 5 min once $O_2$ percentage reached 10%); 5% $CO_2$ (hypercapnia, maintained during 1 min when $CO_2$ percentage reached 5%). Both $O_2$ and $CO_2$ tensions were continuously recorded during the experiments. To calculate changes in respiratory frequency, basal, hypoxic and hypercapnic respiratory frequency was estimated in each animal. Basal respiratory frequency was calculated by averaging the values of 80 digital points (160 s of recording) previous to hypoxia. Peak respiratory frequency during hypoxia was calculated in each animal by averaging the values of 20 points (40 s) at the peak of the hypoxic response. Respiratory frequency during exposure to hypoxia was estimated by averaging either 40 or 150 digital points (80 s or 300 s, respectively) after reaching 10% $O_2$ tension in the chamber before returning to normoxia. Respiratory frequency during exposure to hypercapnia was estimated by averaging 45 digital points (90 s) after reaching ~5% $CO_2$ in the chamber before returning to normoxia[45].

### Measurement of hematocrit and blood lactate
Lactate Plus Meter was used to determine lactate plasma concentration (Nova Biomedical). Control solutions with 1–1.6 and 4–5.4 mM lactate, provided by the supplier, were used to calibrate the device. For measurements, a small drop of blood obtained from the mouse tail after a lancet puncture, was placed on the specific strip for lactate quantification. Hematocrit was measured by collecting blood from an incision of the carotid artery using a hematocrit tube allowing blood to flow by capillarity action into tube. The ends of the tube were sealed with bone wax and the tubes were placed in a microhematocrit centrifuge and span for 5 min at 5.000 rpm. The hematocrit is expressed as the percentage of blood volume occupied by erythrocytes.

### Measurement of body temperature
2–3 months mice were used for body temperature measurements using a rodent thermometer (BIO-TK8851, BIOSeb, In vivo Instruments) to monitor rectal temperature for a few seconds until the measurement was stable and recorded.

### Measurement of insulin-like growth factor-1 in serum
To measure insulin-like growth factor-1 (IGF-1) mice blood was collected in specific MiniCollect ® CAT Serum Tubes (ref. 450533; Greiner Bio One GmbH). After blood collection, the samples were maintained in an upright position for 30 min. The clot was removed by centrifuging at 3.000 × g for 15 min at 4 °C, and the supernatant (serum) was stored at −80 °C. Quantification of insulin-like growth factor (IGF-1) levels in serum samples was done using an ELISA assay (ref. ab240685; Abcam), following the manufacturer's instructions. In both cases, 10 μL serum samples were used in duplicate from each sample. The readout was obtained using a Multiskan Spectrum spectrophotometer (Thermo Electron Corporation). The concentration for each sample was estimated from a reference standard curve with known concentrations of IGF-1.

### Preparation of carotid body slices and dispersed glomus cells
Carotid body (CB) slices were used to measure in glomus cells secretory activity by amperometry and ROS production or cytosolic ATP/ADP ratio by microfluorimetry. These methodologies, as developed in our laboratory, are described in detail elsewhere[26,45,46]. Briefly, CBs dissected from carotid bifurcations were included in 1%(w/v) low

melting point agarose (ref. 50072, FMC BioProducts) in PBS (42 °C). CB sections (150 μm thick) were cut with a vibratome (VT1000S, Leyca) in cold PBS solution and slightly digested with an enzymatic solution (PBS pH 7.4 supplemented with 50 μM CaCl₂, 0.6 mg/ml collagenase II (ref. C6885,Sigma), 0.27 mg/ml trypsin (ref. T8003, Sigma) and 1.25 U/ml porcine elastase (ref. 324682, Millipore) for 5 min at 37 °C. Thereafter, slices were washed with PBS and cultured at 37 °C in a 5% $CO_2$ incubator in DMEM (0 glucose)/DMEM-F-12 (ref. 11966-025/21331-020, Gibco) medium (3:1) supplemented with 100 U/ml penicillin and 10 mg/ml streptomycin (ref. 15140-122, Gibco), 2 mM L-glutamine (ref. 25030-024, Gibco), 10% fetal bovine serum (ref. 10270-106 Gibco), 84 U/L insulin (Actrapid, ref EU/1/02/230/011, Novonordisk), and 1.2 U/ml erythropoietin (ref. EU 1/07/410/028, Sandoz). Slices were used after 24-48 h of incubation.

Dispersed CB glomus cells were used to measure changes in NADH by microfluorimetry. CBs were incubated for 20 min at 37 °C in the same enzymatic solution described for CB slices. Then, CBs were teased apart with needles and incubated for another 5 min at 37 °C. After digestion, cells were mechanically dispersed by pipetting and centrifuged at $300 \times g$. The cell pellet was resuspended in the same culture medium used for slices (without EPO) and plated on glass coverslips treated with poly-L-lysine (ref. P1524, Sigma) and maintained at 37 °C in a 5% $CO_2$ incubator. Dispersed cells were used for experiments after 18–24 h of incubation. Further details of enzymatic dispersion of mouse CB cells are described elsewhere[47].

### Measurement of cytosolic ATP/ADP in carotid body slices

Fluorescence was measured using two-photon laser scanning microscope (2PLSM) containing a multiphoton galvanometer scanning system (Scientifica Ltd) with an Olympus Å-60/1.0 NA water-dipping objective lens. A Chameleon Ultra II (680–1080 nm), 3.5 W Ti: sapphire laser system (Coherent laser group) provided the 2 P excitation source. Laser power attenuation was achieved using two Pockels cell electro-optic modulators (S-MP-4700 and S-MP-6000-INT/UK) in series controlled by Labview. Labview software was written by Scientifica (version 12.0). Non-descanned emission photons were detected with GaAsP photomultiplier tube (PMT), S-MDU-PMT-50, green, 490 nm to 560 nm. PercevalHR, the genetically-encoded ATP/ADP sensor, was expressed in CB cells using an AAV9 vector with TH promoter fragment embedded within it. To estimate the ATP/ADP ratio using PercevalHR, the probe was excited with 950 nm and 820 nm light in rapid succession[10,21]. Green channel (490–560 nm) fluorescent emission signals for both wavelengths were detected using a non-descanned Scientifica S-MDU-PMT-50-50 select GaAsP PMT. Two time series of 5 frames (rate of 3–4 f.p.s., 0.195 Å -0.195 mm pixels) were acquired for each wavelength. Time series analysis was conducted offline using FIJI (version 2.0.0). Multiple cytosolic ROIs and a background ROI were measured, the background was subtracted, and the 950/820 ratio was calculated for each ROI at each time point. The contribution of mitochondria to the bioenergetic status of each cell was estimated by comparing the drop in the PercevalHR ATP/ADP ratio induced by bath application of oligomycin (10 μM) with the drop in the ATP/ADP ratio after perfusion with a modified external solution with oligomycin (10 μM) and glucose (5 mM) replaced with the non-hydrolysable 2-deoxy-glucose (2DG). The OXPHOS index was calculated by dividing the drop in ATP/ADP ratio induced by oligomycin by the sum of total drop in ATP/ADP ratio induced by application of oligomycin and 2DG.

### Amperometric recording of single-cell catecholamine secretion in slices

To monitor single-cell secretory activity we measured by amperometry dopamine secretion from glomus cells in CB slices[46]. Secretory events were recorded with a polarized 10 μm carbon-fiber electrode. An EPC-7 patch-clamp amplifier (HEKA Electronics) was used to measure amperometric currents due to dopamine oxidation. The signal was

filtered at 100 Hz and digitized at 250 Hz before storage on computer. An ITC-16 interface (Instrutech Corporation) and PULSE/PULSEFIT software (version 8.80, HEKA Electronics) were used for data acquisition and analysis. The secretion rate (picocoulombs (pC)/min) was calculated as the amount of charge transferred to the recording electrode during the last minute of exposure to the various stimuli. Basal secretion rate was calculated as the amount of charge measured during a minute in normoxia. For the experiments, a slice was transferred to the recording chamber and continuously perfused with a control extracellular solution containing (in mM): 117 NaCl, 4.5 KCl, 23 NaHCO₃, 1 MgCl₂, 2.5 CaCl₂, 5 glucose and 5 sucrose, at ≈ 35 °C. In 40 mM K⁺ solutions, NaCl was replaced equimolarly with KCl. The "normoxic" solution was bubbled with a gas mixture of 20% $O_2$, 5% $CO_2$, and 75% $N_2$ ($O_2$ tension ≈145 mmHg). The "hypoxic" solution was bubbled with 5% $CO_2$, and 95% $N_2$ to reach an $O_2$ tension of ≈10 to 15 mmHg in the chamber. To do complete dose-response curves (as in Supplementary Fig. 4) solutions with 3% $O_2$, 92% $N_2$ and 6% $O_2$, 89% $N_2$ were also used. With these solutions $O_2$ tensions in the chamber were -30 mmHg and -50 mmHg, respectively[46]. The "hypercapnic" solution was bubbled with 20% $CO_2$, 20% $O_2$ and 60% $N_2$. Osmolality of solutions was ≈300 mOsml/kg and the pH 7.4. Experiments were performed at -35 °C.

### Measurement of NADH and ROS by single-cell microfluorimetry

Microfluorimetric measurements were performed in single dispersed glomus cells (NADH) or cells in CB slices (mitochondrial ROS production)[7,8]. The recording system consists of an inverted microscope (Nikon eclipse Ti) equipped with a 40x/0.60 NA objective, a monochromator (Polychrome V, Till Photonics), and a CCD camera, controlled by Aquacosmos software (version 2.6, Hamamatsu Photonics). For the experiments, dispersed glomus cells plated on poly-L-lysine treated coverslips or CB slices were transferred to the recording chamber and perfused with the solutions described in the preceding section. Experiments were performed at -35 °C. NADH microfluorimetric measurements were performed using a non-ratiometric protocol taking advantage of the NAD(P)H autofluorescence. We tested that the autofluorescence signal mainly reflected changes in NADH because it was rapidly inhibited by the extracellular application of either α-ketobutyrate or pyruvate, which are transported into the cells and rapidly converted to non-metabolizable α-hydroxybutyrate or lactate, respectively[8,20]. NADH was excited at 360 nm and measured at 460 nm. The acquisition protocol was designed with a spatial resolution of 4 × 4 pixels, an excitation time of 150 ms and an acquisition interval of 5 s. A dichroic FF409-Di03 (Semrock) and a band pass filter FF01-510/84 (Semrock) were used. Background fluorescence was subtracted in all experiments. To estimate the rate of change of the NADH signal we used the Igor Pro from WaveMetrics software (version 4.08). The rate of change of the NADH signal was calculated as the increase between the basal and peak autofluorescence amplitudes obtained during exposure to hypoxia divided by the time interval.

Rapid changes in ROS production was performed, in glomus cells in CB slices, using genetically-encoded redox-sensitive green fluorescent protein (roGFP) probes targeted to either the mitochondrial intermembrane space (IMS) or mitochondrial matrix[26,48]. To infect CB slices with the adenoviral vector (ViraQuest Inc), freshly prepared slices were incubated for 24 h in complete culture medium supplemented with $5.5 \times 10^8$ particles/ml IMS-roGFP (VQAd CMV GDP-roGFP), or $6 \times 10^8$ particles/ml matrix-roGFP (VQAd CMV mito-roGFP) to target to specific subcellular compartments. For the experiments the infected CB slice expressing roGFP, was transferred to the recording chamber and bathed with solution (see recording solutions). A dichroic Di02-R488 (Semrock) and a band pass filter FF01-520/35 (Semrock) were used for the experiments. RoGFPs were excited at 400 and 484 nm, and the emission signal was recorded at 535 nm, allowing ratiometric measurements of rapid and reversible changes in cellular

redox state. To ensure that the probes were working correctly, in most experiments a brief pulse of $H_2O_2$ (0.1 mM) was added at the end to measure the cell maximum oxidation signal.

## Measurement of cytosolic $Ca^{2+}$ by single-cell microfluorimetry

Microfluorimetric measurements of intracellular changes in $Ca^{2+}$ concentration were performed in single dispersed glomus cells[5,7,47]. Dispersed CB cells were loaded with Fura2- AM (4 mM Fura2- AM; TefLabsMW1002, in DMEM/F-12 without serum) at 37 °C for 30 min and subsequently incubated for 15 min in complete medium to remove excess Fura2-AM. After loading cells, the coverslip was transferred to the recording chamber and continuously perfused with external solution. The set up consists of an inverted microscope (Nikon eclipse Ti) equipped with a 40x/0.60 NA objective, a monochromator (Polychrome V, Till Photonics), and a CCD camera, controlled by Aquacosmos software (version 2.6, Hamamatsu Photonics). Alternating excitation wavelengths of 340 and 380 nm, and an emission wavelength of 510 nm were used. Background fluorescence was subtracted before obtaining the F340/F380 ratio. A dichroic FF409-Di03 (Semrock) and a band pass filter FF01-510/84 (Semrock) were used. Cytosolic $[Ca^{2+}]$ signals were digitized at a sampling interval of 500 ms.

## Real-time quantitative PCR

Total RNA was isolated from the CB and superior cervical ganglion (SCG) using RNeasy Micro Kit (ref. 74004, Qiagen) or from kidney using Trizol (ref. 15596026, Life Technologies) following manufacturer's instructions. Each SCG replicate was isolated from 1 or 3 mice. Each CB replicate was pooled from 4–5 mice to obtain enough amount of RNA. Complementary RNA (cRNA) was then amplified from CB RNA using GeneChip WT PLUS Reagent Kit (ref. 902622, Affymetrix). RNA from the SCG and kidney and cRNA from the CB were copied to cDNA using QuantiTect Reverse Transcription Kit (ref. 205311, Qiagen). Real-time quantitative PCR reactions were performed in a 7500 Fast Real-Time PCR System (Applied Biosystem) using Taqman Gene Expression Assays for *Ndufs2*, *Sdhd* and *NDI1* (Thermo Fisher Scientific). *Ppia* (and *Gapdh*) were also estimated in each sample to normalize the amount of RNA or cRNA input in order to perform relative quantification.

## Mitochondrial complex I activity

Mitochondrial complex I (MCI) activity in the SCG and kidney was determined using complex I enzyme activity dipstick assay kit (ref. ab109720, Abcam) following the manufacturer's instruction with minor modifications[7]. For each replicate, SCGs pooled from 5 mice were homogenized in 100 µl extraction buffer followed by incubation on ice for 20 min. The homogenate was then centrifuged at $16,000 \times g$ for 30 min. The supernatant was collected for the enzymatic assay. Forty mg of kidney from each mouse was homogenized in 200 µl extraction buffer to prepare homogenate. Eight (SCG) or 5 (kidney) µg of protein extract, which was determined using the Bio-Rad protein assay (Bio-Rad), were applied to measure the MCI activity. MCI was immunocaptured on the dipstick, which was immersed in a solution containing NADH as a substrate and nitrotetrazolium blue (NBT) as the electron acceptor. Immunocaptured MCI oxidized NADH and reduced NBT to form a blue-purple precipitate. The signal intensity of this precipitate, which corresponds to the level of MCI activity, was scanned with an image analyzer (ImageQuant LAS 4000mini, GE Healthcare). The signal intensity was quantified using the ImageQuant TL software (GE Healthcare).

## Immunohistochemical analysis

For immunofluorescent studies, mice were first perfused with PBS, then with 4% paraformaldehyde in PBS before tissue dissection. Dissected carotid bifurcation was fixed with 4% paraformaldehyde in PBS for 2 h, cryoprotected overnight with 30% sucrose in PBS, and embedded in OCT (ref. 4583, Tissue-Tek). Tissue sections of 8 µm were cut with a cryostat (Leica). Tissue sections were incubated with primary antibodies overnight at 4 °C: TH (1:2500 dilution, NB300-109, Novus Biological Inc.), TH (1:200 dilution, AB1542, Millipore), NDUFS2 (1:200 dilution, ab192022, Abcam), or GFP (1:400 dilution, 1010, Aves Labs Inc.). This was followed by incubation with fluorescent secondary antibodies (1:500 diluiton): Alexa Fluor™ 488: Donkey anti-Sheep IgG (H + L) Cross-Adsorbed Secondary Antibody, Invitrogen (A11015) and Alexa Fluor™ 568: Goat anti-Rabbit IgG (H + L) Cross-Adsorbed Secondary Antibody, Invitrogen (A11011). Nuclei were labeled with 4′,6′-diamidino-2-phenylindole (DAPI). Immunofluorescent images were obtained using Leica Stellaris 8 confocal microscopy (Leica) and Leica Application Suite X software (Leica Microsystems).

## Quantification and statistical analysis

Statistical analyses were carried out using Prism Version 8.2.1 (279) for MacOS. Normality of the data sets was tested with the Shapiro-Wilk test, D'Agostino and Pearson test, or Kolmogorov-Smirnov test. In cases, a log transformation was performed to normalize the distribution prior to parametric analysis. Data with normal distribution were described as mean±standard error of the mean (SEM) with the number (n) of experiments indicated. For graphical representation of the data with normal distribution we used bar diagrams, with indication of the mean ± SEM, and a scatter plot of the data points superimposed. For graphical representation of non-parametric data, we used boxplots with indication of median, quartiles, and whiskers (highest and lowest values). Comparison between two groups was done as follows: For parametric data we used paired or unpaired t tests, depending on the experiments. For non-parametric data we used Mann–Whitney unpaired t test. All t tests, parametric and non-parametric, were two-tailed. Comparison of several groups was done as follows: For parametric data we used one-way ANOVA followed by Tukey's multiple comparisons test or Newman-Keuls multiple comparisons test. In both cases we compared every group with each other. Dunnett's multiple comparisons test was used to compare two groups with respect to a control group. For non-parametric data, we used the Dunn's multiple comparisons test to compare every group with each other. Statistical tests used are indicated in the figure legends. Statistically significant *P* values (<0.05) are represented in the figures.

## Ethics approval

This study was performed following the recommendations set out in the Global Code of Conduct for Research in Resource-Poor Settings.

## Reporting summary

Further information on research design is available in the Nature Portfolio Reporting Summary linked to this article.

# Data availability

Data generated or analyzed over the course of this study are included within the paper. The data that support the findings of this study are available from the corresponding author upon request. Source data are provided with this paper.

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

## Acknowledgements

We thank staff of IBiS and "Centro de Producción y Experimentación Animal, Oscar Pintado" for technical assistance. We also than Dr. Alejandro Moreno-Domínguez and Mr. Ignacio Arias-Mayenco for help in setting up the two-photon microscope. We are grateful to Dr. James Surmeier for generous supply of TH-driven Perceval plasmid. This research was supported by the Andalusian Government (FEDER Andalucía 2014–2020, 2018 Call, US-1255654, P.O.-S. and L.G.), the Spanish Ministries of Science and Innovation and Health (Grants SAF2016-74990-R and PID2019-106410RB-I00 funded by MCIN/AEI/10.13039/501100011033, J.L.-B., L.G. and P.O.-S.), and the European Research Council (ERC Advanced Grant PRJ201502629, J.L.-B.). B.J.-G. received a predoctoral fellowship (FPI program) from the Spanish Government.

## Author contributions

B.J.-G. P.O.-S., L.G., P.G.-R. and P.G.-F. performed the experiments and analyzed the data. N.C. provided essential material for the experiments. All authors contributed to the design of the study and to generating the draft of the paper. J.L.-B. coordinated the project and the writing of the paper.

## Competing interests

The authors declare no competing interests.
