## [Peer Review File · Nature Communications]

Transgenic NADH dehydrogenase restores oxygen regulation of breathing in mitochondrial complex I-deficient miceREVIEWER COMMENTS

Reviewer #1 (Remarks to the Author):

This paper investigates the role of mitochondrial complex I in acute O₂ sensing in peripheral chemoreceptors and isolated chemoreceptor cells. The results provide an original hypothesis specifically linking NADH dehydrogenase activity of MCI (rather than proton pumping) to O₂ sensing in this dedicated system. Increased NADH production and ROS production in the inter membrane space appears as key elements triggering the hypoxic response. This is an important contribution to the field, building and critically expanding previous works showing the key role of MCI for O₂ sensing.

The experiments have been conducted using mice with deletion of NDUFS2 directed to TH⁺ neurons. NDUFS2 is a subunit necessary for MCI assembly and function, and while the deletion is by design restricted to TH⁺ neurons, these mice are severely growth restricted, with low IGF1, high blood lactate, and have an extremely short life span that do not extend beyond 2 to 3 months. They are also characterized by lack of proper functions of peripheral chemoreceptors evidenced by absence of hypoxic ventilatory response in vivo, total disruption of secretory, NADH and ROS hypoxic responses of type I cells in vitro. Partial recovery of MCI function has been achieved by breeding the MCI KO mice to express the yeast protein NDI1 that is anchored to the matrix side of the mitochondrial membrane, allowing oxidation of NADH and thus re-establishing normal convergent electron flux through CoQ by CI and CII and downstream CIII/CIV activity. However, proton pumping by CI is not achieved in these mice, but remarkably there is a full recovery of growth, in vivo and in vitro hypoxic responses. The data set also includes measurements of ROS production with unprecedented details showing that hypoxia specifically increases inter-membrane space ROS levels, while it reduces matrix ROS production.

An extended data set supports the main findings by providing immunohistochemical, RT-PCR and functional (MCI activity) evidence of the NUDFS2 deletion in carotid bodies and superior cervical ganglion. Additional experiments are also presented, including a line of WT mice with inclusion of NDI1 on top of a functional MCI and another model of MCI disruption with a tamoxifen inducible KO with or without NDI1 recovery. Results with this last line concerning the role of NADH oxidation on O₂ sensing of peripheral chemoreceptor cells are similar than with the original KO, showing total disruption and recovery with NDI1. They furthermore add highly intriguing data suggesting that the yeast protein NDI1 is a more efficient O₂ sensor than the full mammalian MCI.

Overall, this is an impressive set of data that strongly supports the leading hypothesis of the authors and provide original and important data to further our understanding of the mechanisms involved in O₂ sensing in peripheral chemoreceptors. Given the data presented, the conclusions reached by the authors are sound and logical, and there are no apparent flaws that would invalidate the work.

I see however the following point of weaknesses:

- No experiments have been performed to evaluate the respective roles of NADH vs ROS as immediate downstream mechanisms linking MCI to cellular hypoxic responses in WT or KO+NDI1 mice. Giving that both responses are exaggerated in the rescued KO mice compared to WT, adding these experiments would have been an additional step forward of potential interest.
- What happens to the TCA cycle, ETC functions and OXPHOS with MCI deletion and NDI1 recovery? To which extent does the lack of proton pumping by NDI1 limit OXPHOS capacity and ATP synthesis? It seems ok to conclude that O₂ sensing is selectively independent of CI proton pumping, but would it be possible that SDH/CII contribution to ETC is increased with rescue of NADH oxidation, providing full recovery of proton pumping by CIII/CIV and downstream OXPHOS/ATP synthesis? The uncertainty should be recognized unless the authors are able to provide data or references showing otherwise.
- It is not clear if the authors intend to draw links between altered chemoreceptor functions, growth

retardation and short life span. A statement clearly identifying whether it is reasonable to draw this link could be useful, or are there any other known/suspected cause of dwarfism and short life span in these mice?

- For further thoughts, since mice lacking functional central chemoreceptors die within the first postnatal hours (see Phox2b27Ala/+ mice in PMID 34478357), it is certainly impressive that mice without functional O₂ sensing in peripheral chemoreceptors can survive up to 75 days. Yet, it is very clear that ventilatory support by peripheral chemoreceptors is vital during early postnatal life as shown by experiments with peripheral chemodeneration in rat pups. Can we hypothesize that MCI role in O₂ sensing or ETC functions evolves during postnatal development? Could you briefly comment? Have you been able to evaluate pre-weaning mortality in these mice?

- Data showing high hematocrit values after 10 days of chronic hypoxia in KO mice and normalization with NDI1 expression are presented as being equivalent to an altered ventilatory acclimatization to hypoxia. However, and while I agree that hematological and ventilatory acclimatization are linked, in absence of other respiratory parameters this is slightly misleading. These data do not add much relevant information and should be deleted to avoid potential confusion.

--

opting into transparent peer review scheme, review by V. Joseph, U Laval (Qc - Canada)

Reviewer #3 (Remarks to the Author):

Fully understanding oxygen-sensing molecular mechanisms in acute responses to hypoxia, and in particular, hypoxic ventilatory response driven by carotid body glomus cells, is still a challenge nowadays. The group of Prof. Lopez-Barneo has done several major contributions in this field, including the discovery of mitochondrial complex I (MCI) being a key player in this acute oxygen-sensing mechanism (ref. 7, 2015). However, several questions remain unclear regarding how MCI contributes to this molecular mechanism, taking into account that MCI is a complicated molecular machine that carries at the same time different functions: it transport electrons between NADH and CoQ, it pumps protons and it produces superoxide. Indeed, another important question is that if MCI could be a primary oxygen sensor by itself.

In this paper, Jimenez-Gomez et al. address successfully some of those questions, by studying mice that are KO for a MCI subunit in TH-expressing cells (neurons, including the glomus cell of carotid body), the same model used in ref. 7, but now comparing with the TH promoter-driven expression of NDI1, a yeast protein, structurally very different from MCI, that can substitute MCI in the NADH:CoQ oxidoreductase function, but not in the proton pumping; NDI1 can also produce superoxide, but with different mechanisms than MCI.

The mouse model is well confirmed, and the experiments in general are clearly explained and well performed, providing a clear picture that NDI1 expression can recover hypoxia sensitivity in glomus cells. With these experiments, the authors clearly show that NADH:CoQ oxidoreductase seems to be fundamental for the oxygen sensing mechanism and that complete MCI is not needed for this activity, so it does not seem to be the primary oxygen sensor.

Indeed, as the authors state, the experiments also show that the response in NDI1-expressing cells (not only KO cells + NDI1, but also WT cells + NDI1) is different in some aspects than in the normal WT cells, which will help to discriminate downstream details of the mechanism, such as the relative importance of NADH and ROS variations in future works.

Taking into account the major result that NDI1 can restore hypoxia sensitivity, as well as the observed differences when NDI1 is present, the authors propose a model for MCI role in acute oxygen sensing in glomus cells. However, this part is weaker and some parts of the model and of the conclusions they provide are not so clear, as described below.

Additionally, the authors show some interesting features of the phenotype of animals expressing NDI1 that rescue other consequences of the previous KO model.

The paper is very interesting for being published in Nature Communications, although some

improvements would be needed to strengthen some of the results shown, as well as trying to answer some questions that arise and could reinforce the major conclusions or solve some doubts that are present in the manuscript.

Major points:

1.a. Experiments in Extended Data (ED) Figs. 11 and 12 seem to be preliminary. In order to present them, they would be better shown with adequate quantification, as in the rest of the paper.

b. In ED Fig. 12, a comparison with ESR-WT and ESR-KO animals would be needed, to see the magnitude of the response.

c. What would be the explanation for the rotenone effect in ED Fig 12b? (And why is there a decreasing slope at the start of the first Hx treatment in the same ED Fig 12?).

d. ROS measurement in ESR-WT, ESR-KO and ESR-KO/NDI1 would be very interesting to provide the whole picture.

2. A recent paper by Hernansanz-Agustin et al. (PMID 32728214) provided a model for the molecular mechanism driving redox acute response to hypoxia that could fit with the one provided here. Even though the glomus cell can be a specialized cell with differences, some aspects of this model would be worthy to test in the NDI1 expression model (and feasible to measure with adequate fluorescent probes):

a. Slowing down or reversal of MCI would produce matrix acidification due to less proton pumping, but that would not be the case for NDI1 (unless proton pumping is also abolished in complexes III and IV). Is mitochondrial matrix being acidified in response to hypoxia in the KO-NDI1 and WT-NDI1 cells, compared with WT or KO cells?

b. According to this model, a downstream effect of matrix acidification is reduction of inner mitochondrial membrane diffusion due to increased mitochondrial Na⁺ import. Is the inner mitochondrial membrane diffusion being slowed down in response to hypoxia in the KO-NDI1 and WT-NDI1 cells?

3. Some parts of the model described (Fig. 4 i,j) are not easily understood or well supported.

a. It is strange that in WT cells MCI produce ROS directly to the IMS. Most models of superoxide production by MCI locate it directed to the matrix, while IMS superoxide production is driven mainly by the Qo site of complex III.

b. In the NDI1 cells, why would the matrix dehydrogenases be less active and produce less ROS? There is matrix NADH accumulation and ROS increase, so they could be more active and producing more superoxide.

c. In the NDI cells, why are IMS ROS assumed to come from the matrix? They could be produced directly to the IMS by complex III, for example, even in less extent than in the WT cells.

4. The paper assumes that the presence of NDI1 overcomes NADH:CoQ activity of MCI. It is an interesting suggestion, but in that case the proton gradient and OXPHOS function could also be affected. It could be worthy trying to confirm this point by measuring the proton gradient or the mitochondrial membrane potential, or any other way to assess specifically the function of MCI.

5. Another assumption of the paper is that NDI1 is reversed or slowed down during hypoxia. Could this be assessed experimentally? Maybe using NDI1 inhibitors, or forcing reversal in normoxia providing an excess of succinate...

6. Also interesting is the fact that NDI1 expression affects CO₂ sensing in both KO/NDI1 and WT/NDI1 cells. In p. 8 the authors suggest an explanation based on the Na⁺/H⁺ antiporter activity of inactive MCI. Would this activity of MCI be maintained in the KO/NDI1 cells?

Minor points:

p.3. The *Ndufs2* KO mice are described as (*Ndufs2* flox^{-/-};Cre), are they homozygous or heterozygous for *Ndufs2* flox and Cre-dependent deletion?

p.4 and ED Fig. 2c & 9g. Please describe what type of activity of MCI is measured, as it is supposed to be different from the NADH:CoQ oxidoreductase activity that is also exerted by NDI1.

p. 4 explains Fig. 1f as “hypoxia-induced lactatemia”, but the figure is described as basal blood lactate.

p. 6 and Fig. 4 c,g. It would be good to say in the text that the IMS ROS signal in KO-ND11 is lower than in WT cells (the scale in Fig. 4c is different than in Fig. 4a).

Fig. 4i is not explained in the legend.

ED Fig. 8c. It would be better to show the median, as in Fig. 3j. Also ED Fig 8h,j would be better in the same scale and graph type as in Fig. 4g,h.

Reviewer #4 (Remarks to the Author):

The study by Jimenez-Gomez and colleagues extends their previous work examining the function of MCI and its NADH dehydrogenase activity on acute O₂ sensing of the carotid body glomus cells. They show, as they have previously, that knockout mice lacking MCI specifically in TH⁺ cells have a nearly absent hypoxic response which is also associated with abolished dopaminergic neurosecretion (by CB glomus cells in vitro), NADH and ROS accumulation. All of these effects are reversed in transgenic mice that express a yeast NADH dehydrogenase (ND1) driven by the TH promoter. While the data are sound, I have a number of serious concerns that should be addressed.

1. The authors show absent hypoxic responses in KO mice, with restoration in ND1 Tg mice. ND1 is expressed in all TH⁺ cells, so how can the authors be sure that the restoration of the HVR is specific to the glomus cells? Indeed, catecholaminergic neurons are distributed throughout the neural circuitry involved in the HVR, including the NTS, pons, VLM etc. Did the authors consider an effect at these other sites?

2. Related to concern 1 above. It would be more convincing to show effects of KO and ND1 rescue at the level of the CSN afferents. As it stands, the only data shown related to glomus cell response is dopamine secretion. However, these data are from isolated glomus cells that are exposed to PO₂ of 10-15mmHg (i.e. not physiological). Moreover, it is apparent in their own data that neurosecretion in vitro may correlate poorly with the strength of the conscious animal (compare Suppl. Figs 6 (whole animal response of WT/ND1 mice) with Suppl. 7 (secretory response from WT/ND1 glomus cells)).

3. As the KO is across all TH⁺ neurons, are the authors sure that the decreased HVR may be related to altered metabolic drive? I ask because Fig. 1d and Suppl. Fig 10 suggest reduced respiratory frequency at baseline, suggesting reduced drive. Are there any differences in body temperature between the WT, KO and KO+ND1 mice?

4. Fig. 1b suggests that the KO/ND1 mice have a very transient HVR (over about 3 time bins or ~20 seconds). Was this analyzed, and how come it is not reflected in the data shown in Suppl. Fig. 3 (peak response).

Minor:

1. It would be nice to see actual evidence of ND1 expression, in addition to GFP immunofluorescence.

2. Some of the figures have data for "Sdhd", but I could not find a definition for this molecule.

RESPONSE TO THE REVIEWER'S COMMENTS
Jiménez-Gómez et al. **Authors response in red.**

Reviewer #1 (Remarks to the Author)

This paper investigates the role of mitochondrial complex I in acute O₂ sensing in peripheral chemoreceptors and isolated chemoreceptor cells. The results provide an original hypothesis specifically linking NADH dehydrogenase activity of MCI (rather than proton pumping) to O₂ sensing in this dedicated system. Increased NADH production and ROS production in the inter membrane space appears as key elements triggering the hypoxic response. This is an important contribution to the field, building and critically expanding previous works showing the key role of MCI for O₂ sensing.

The experiments have been conducted using mice with deletion of NDUFS2 directed to TH⁺ neurons. NDUFS2 is a subunit necessary for MCI assembly and function, and while the deletion is by design restricted to TH⁺ neurons, these mice are severely growth restricted, with low IGF1, high blood lactate, and have an extremely short live span that do not extend beyond 2 to 3 months. They are also characterized by lack of proper functions of peripheral chemoreceptors evidenced by absence of hypoxic ventilatory response in vivo, total disruption of secretory, NADH and ROS hypoxic responses of type I cells in vitro. Partial recovery of MCI function has been achieved by breeding the MCI KO mice to express the yeast protein NDI1 that is anchored to the matrix side of the mitochondrial membrane, allowing oxidation of NADH and thus re-establishing normal convergent electron flux through CoQ by CI and CII and downstream CIII/CIV activity. However, proton pumping by CI is not achieved in these mice, but remarkably there is a full recovery of growth, in vivo and in vitro hypoxic responses. The data set also includes measurements of ROS production with unprecedented details showing that hypoxia specifically increases inter-membrane space ROS levels, while it reduces matrix ROS production.

An extended data set supports the main findings by providing immunohistochemical, RT-PCR and functional (MCI activity) evidence of the NDUFS2 deletion in carotid bodies and superior cervical ganglion. Additional experiments are also presented, including a line of WT mice with inclusion of NDI1 on top of a functional MCI and another model of MCI disruption with a tamoxifen inducible KO with or without NDI1 recovery. Results with this last line concerning the role of NADH oxidation on O₂ sensing of peripheral chemoreceptor cells are similar than with the original KO, showing total disruption and recovery with NDI1. They furthermore add highly intriguing data suggesting that the yeast protein NDI1 is a more efficient O₂ sensor than the full mammalian MCI.

Overall, this is an impressive set of data that strongly supports the leading hypothesis of the authors and provide original and important data to further our understanding of the mechanisms involved in O₂ sensing in peripheral chemoreceptors. Given the data presented, the conclusions reached by the authors are sound and logical, and there are no apparent flaws that would invalidate the work.

Authors: We thank the reviewer for the positive comments on our work. We also thank him/her for the time spent in a detailed evaluation of our manuscript and the insightful suggestions, which have helped us to improve the quality of the work. We have performed new experiments and added new data and explanations to the revised manuscript to address all the questions raised by the reviewer. With the new data the manuscript has been expanded and adapted to the format of Nature Communications. We have also added a new author (Dr. Patricia González-Rodríguez), who collaborated in the generation and characterization of the conditional Ndufs2 KO mice. More recently, she has also contributed

to the project in the study of ATP levels in glomus cells using two-photon microscopy and a genetically encoded fluorescent probe.

I see however the following point of weaknesses:

- No experiments have been performed to evaluate the respective roles of NADH vs ROS as immediate downstream mechanisms linking MCI to cellular hypoxic responses in WT or KO+NDI1 mice. Given that both responses are exaggerated in the rescued KO mice compared to WT, adding these experiments would have been an additional step forward of potential interest.

Authors: The role of NADH and H₂O₂ in the regulation of membrane ionic currents and electrical responses to hypoxia in glomus cells was described in our original paper in which WT and KO (MCI-deficient) mice were studied (Figs. 3, 6 and Supplementary Figs. 2 and 7 in Fernandez-Agüera et al., ref. 7). More recently, we have also shown that pharmacological downregulation of NADH levels in the cytosol inhibits the secretory response to hypoxia of glomus cells (Fig.4g in Cabello-Rivera et al., 2022, ref. 17). Given that the electrophysiological experiments require a high number of mice we did not find it necessary to repeat them in KO/NDI1 mice. The fact that hypoxia-induced single cell secretory activity (in an external Ca²⁺-dependent manner) is rescued by NDI1 expression (Fig. 3e) clearly demonstrates that NDI1 can replace MCI for acute O₂ sensing in glomus cells. However, to further support this point we have done dose-response secretory experiments (Supplementary Fig. 4; page 6, last 4 lines and page 7 lines 1-2) and cytosolic Ca²⁺ measurements (Supplementary Fig. 5; page 6, last 4 lines and page 7 lines 1-2) to compare the downstream hypoxic responses (cytosolic Ca²⁺ and secretory activity) in WT and KO/NDI1 glomus cells.

Regarding the amplitude of the mitochondrial responses to hypoxia, we see an increase in the average amplitude of the NADH autofluorescence signal in KO/NDI1 mice in comparison with WT mice (Fig. 4j). However, the intermembrane space ROS signals are not too different (Fig.5g). Although there is a large variability in the data, there is a trend for the secretory response to hypoxia to be higher in KO/NDI1 cells respecting WT glomus cells (Fig. 3j).

- What happens to the TCA cycle, ETC functions and OXPHOS with MCI deletion and NDI1 recovery? To which extent does the lack of proton pumping by NDI1 limit OXPHOS capacity and ATP synthesis? It seems ok to conclude that O₂ sensing is selectively independent of CI proton pumping, but would it be possible that SDH/CII contribution to ETC is increased with rescue of NADH oxidation, providing full recovery of proton pumping by CIII/CIV and downstream OXPHOS/ATP synthesis? The uncertainty should be recognized unless the authors are able to provide data or references showing otherwise.

Authors: To address this question of the reviewer we have done ratiometric fluorescent measurements of intracellular ATP/ADP ratio using a genetic probe (Perceval; Tantama et al., ref. 21) and a two-photon microscope. These new data indicate that as it occurs in other neuronal cell types (Gonzalez-Rodriguez et al., ref. 10), MCI-deficient glomus cells seem to utilize glycolytic ATP to reverse MCV function and to maintain mitochondrial membrane potential. Oxidative ATP synthesis is recovered in KO/NDI1 cells. As suggested by the reviewer, it seems that restoration of metabolism in MCI-deficient cells by NDI1 expression allows proton pumping by MCIII and MCIV to compensate for the lack of MCI. These data are included in Fig. 2e,f and described in page 5 (last paragraph) and page 6 (first paragraph) as well as in the Method section (page 15).

- It is not clear if the authors intend to draw links between altered chemoreceptor functions, growth retardation and short life span. A statement clearly identifying whether it is reasonable to draw this link could be useful, or are there any other known/suspected cause of dwarfism and short life span in these mice?

Authors: We think that there are no links between altered chemoreceptor function in KO mice and their growth retardation and short life span. These are systemic effects secondary to dysfunction of MCI in other cell types unrelated to acute O₂ sensing in which the TH promoter is active. We have modified the text to clarify this point (page 5, lines 7-10).

- For further thoughts, since mice lacking functional central chemoreceptors die within the first postnatal hours (see Phox2b27Ala/+ mice in PMID 34478357), it is certainly impressive that mice without functional O₂ sensing in peripheral chemoreceptors can survive up to 75 days. Yet, it is very clear that ventilatory support by peripheral chemoreceptors is vital during early postnatal life as shown by experiments with peripheral chemodenervation in rat pups. Can we hypothesize that MCI role in O₂ sensing or ETC functions evolves during postnatal development? Could you briefly comment? Have you been able to evaluate pre-weaning mortality in these mice?

Authors: To address the reviewer comment we have added a chart (Fig. 1g) illustrating that there is no an appreciable increase in pre-weaning mortality in KO mice respecting WT mice. We think that this is due to the fact that although Cre-mediated recombination of the Ndufs2 allele may occur in the last days before birth, wash out of mitochondrial proteins takes several weeks (Fornasiero et al., ref. 18). This is the reason why mortality of KO mice (Fig. 1g) and decrease in animal weight (Supplementary Fig. 1e in Fernandez-Aguera et al., ref. 7; Supplementary Fig. 2a-c) become appreciable only 3-4 weeks after birth. A similar delay in the appearance of the MCI-deficient phenotype has been described in a recent study in central dopaminergic neurons using the same conditional Ndufs2 floxed MCI-deficient mice (Gonzalez-Rodriguez et al., 2021, ref. 10). Please see comment on page 5, lines 4-7. We have confirmed in neonatal rats that as it occurs in adult rats and mice (Fig. 2e in Ortega-Saenz et al., ref. 22; Fig. 6 in Garcia-Fernandez et al., 2007, PMID:17827405; Fig. 3 in the current manuscript) sensitivity to hypoxia is occluded by rotenone, a selective MCI blocker. It was this effect of rotenone on glomus cells from adult rodents that led us to initiate our current investigation on the role of mitochondria in acute O₂ sensing. We show in Fig. R1 a representative experiment only for reviewing purposes.

Figure R1. Amperometric recording of secretory activity induced by hypoxia in a glomus cell in a carotid body slice (5 days-old rat). Similar to hypoxia, rotenone activates cell secretion but no further effect of hypoxia is observed in the presence of rotenone.

- Data showing high hematocrit values after 10 days of chronic hypoxia in KO mice and normalization with NDI1 expression are presented as being equivalent to an altered ventilatory acclimatization to hypoxia. However, and while I agree that hematological and ventilatory acclimatization are linked, in absence of other respiratory parameters this is slightly misleading. These data do not add much relevant information and should be deleted to avoid potential confusion.

Authors: Following the suggestion of the reviewer the hematocrit figure and any reference to acclimatization has been removed. However, we think that the hematocrit data by itself is

interesting as another manifestation in KO mice that is rescued by NDI1 expression. These data now appear in Supplementary Fig. 2e.

Reviewer #3 (Remarks to the Author)

Fully understanding oxygen-sensing molecular mechanisms in acute responses to hypoxia, and in particular, hypoxic ventilatory response driven by carotid body glomus cells, is still a challenge nowadays. The group of Prof. Lopez-Barneo has done several major contributions in this field, including the discovery of mitochondrial complex I (MCI) being a key player in this acute oxygen-sensing mechanism (ref. 7, 2015). However, several questions remain unclear regarding how MCI contributes to this molecular mechanism, taking into account that MCI is a complicated molecular machine that carries at the same time different functions: it transport electrons between NADH and CoQ, it pumps protons and it produces superoxide. Indeed, another important question is that if MCI could be a primary oxygen sensor by itself. In this paper, Jimenez-Gomez et al. address successfully some of those questions, by studying mice that are KO for a MCI subunit in TH-expressing cells (neurons, including the glomus cell of carotid body), the same model used in ref. 7, but now comparing with the TH promoter-driven expression of NDI1, a yeast protein, structurally very different from MCI, that can substitute MCI in the NADH:CoQ oxidoreductase function, but not in the proton pumping; NDI1 can also produce superoxide, but with different mechanisms than MCI. The mouse model is well confirmed, and the experiments in general are clearly explained and well performed, providing a clear picture that NDI1 expression can recover hypoxia sensitivity in glomus cells. With these experiments, the authors clearly show that NADH:CoQ oxidoreductase seems to be fundamental for the oxygen sensing mechanism and that complete MCI is not needed for this activity, so it does not seem to be the primary oxygen sensor.

Indeed, as the authors state, the experiments also show that the response in NDI1-expressing cells (not only KO cells + NDI1, but also WT cells + NDI1) is different in some aspects than in the normal WT cells, which will help to discriminate downstream details of the mechanism, such as the relative importance of NADH and ROS variations in future works. Taking into account the major result that NDI1 can restore hypoxia sensitivity, as well as the observed differences when NDI1 is present, the authors propose a model for MCI role in acute oxygen sensing in glomus cells. However, this part is weaker and some parts of the model and of the conclusions they provide are not so clear, as described below. Additionally, the authors show some interesting features of the phenotype of animals expressing NDI1 that rescue other consequences of the previous KO model. The paper is very interesting for being published in Nature Communications, although some improvements would be needed to strengthen some of the results shown, as well as trying to answer some questions that arise and could reinforce the major conclusions or solve some doubts that are present in the manuscript.

Authors: We thank the reviewer for the positive comments on our work. We also thank him/her for the time spent in a detailed evaluation of our manuscript and the insightful suggestions, which have helped us to improve the quality of our work. We have performed new experiments and added new data and explanations to the revised manuscript to address all the questions raised by the reviewer. With the new data the manuscript has been expanded and adapted to the format of Nature Communications. We have also added a new author (Dr. Patricia González-Rodríguez), who collaborated in the generation and characterization of the conditional *Ndufs2* KO mice. More recently, she has also contributed to the project in the study of ATP levels in glomus cells using two-photon confocal microscopy and a genetically encoded fluorescent probe.

Major points:

1.a. Experiments in Extended Data (ED) Figs. 11 and 12 seem to be preliminary. In order to

present them, they would be better shown with adequate quantification, as in the rest of the paper.

Authors: We have done new experiments and completed the data presented in previous ED Figure 11 as suggested by the reviewer (Fig. 7c-m in the revised manuscript). We have also completed previous ED Fig. 12 (now Supplementary Fig. 11).

b. In ED Fig. 12, a comparison with ESR-WT and ESR-KO animals would be needed, to see the magnitude of the response.

Authors: Comparison between the NADH signals between ESR-WT and ESR-KO mice was already published in a previous paper of our laboratory (Fig. 2 a-d in Arias-Mayenco et al., ref. 8). In Supplementary Fig. 11 we now show new data from control mice (ESR-WT) in which it is clearly seen that rotenone increases NADH levels and occludes any further effect of hypoxia (panel a). In ESR-KO/NDI1 mice rotenone has no (or very little) effect whereas hypoxia produces a clear and reversible increase in NADH in the presence of rotenone (panel b).

c. What would be the explanation for the rotenone effect in ED Fig 12b? (And why is there a decreasing slope at the start of the first Hx treatment in the same ED Fig 12?).

Authors: In some cells (~25%) of the ESR-KO/NDI1 mouse model rotenone produced a small increase in NADH probably because these cells still have some functional MCI. Regarding the slow changes in the recordings, we think they are due to photobleaching of the fluorescent molecule. We now present a revised figure (Supplementary Fig. 11) that includes new data and the quantification requested by the reviewer.

d. ROS measurement in ESR-WT, ESR-KO and ESR-KO/NDI1 would be very interesting to provide the whole picture.

Authors: A detailed comparison between the mitochondrial ROS signals (matrix and intermembrane space) in glomus cells from ESR-WT and ESR-KO mice was already published in a previous paper of our laboratory (Figs. 2 and 3, Suppl. Fig. 4 in Arias-Mayenco et al., ref. 8). Recovery of the ROS signal by NDI1 expression has been studied in detail in KO/NDI1 mice. It was not studied in the ESR-KO/NDI1 model because these experiments are particularly difficult to perform and require the use of many Tamoxifen-treated animals. The mice available have been used to do the quantitative study of the hypoxia-induced glomus cells secretory activity (Fig. 7) and of the NADH signal indicated by the reviewer (Supplementary Fig. 11). We think that the rescue of glomus cells O₂ sensing in the adult conditional MCI-deficient mice expressing NDI1 is sufficiently well demonstrated with the secretory (Fig. 7) and NADH (Supplementary Fig. 11) experiments. These data are completed with the recovery of the hypoxic ventilatory response shown in Fig. 6.

2. A recent paper by Hernansanz-Agustin et al. (PMID 32728214) provided a model for the molecular mechanism driving redox acute response to hypoxia that could fit with the one provided here. Even though the glomus cell can be a specialized cell with differences, some aspects of this model would be worthy to test in the NDI1 expression model (and feasible to measure with adequate fluorescent probes):

a. Slowing down or reversal of MCI would produce matrix acidification due to less proton pumping, but that would not be the case for NDI1 (unless proton pumping is also abolished in complexes III and IV). Is mitochondrial matrix being acidified in response to hypoxia in the KO-NDI1 and WT-NDI1 cells, compared with WT or KO cells?

b. According to this model, a downstream effect of matrix acidification is reduction of inner mitochondrial membrane diffusion due to increased mitochondrial Na⁺ import. Is the inner

mitochondrial membrane diffusion being slowed down in response to hypoxia in the KO-NDI1 and WT-NDI1 cells?

Authors: A publication by Hernansanz-Agustin et al. cited in the original manuscript (ref. 30) reported that in some cell classes hypoxia induces an increase in ROS due to the transition of MCI to a deactive state. In Fernandez-Aguera et al. (ref. 7), we discussed in detail that the fast and highly reversible ROS and NADH signals recorded in glomus cells exposed to hypoxia are not compatible with MCI deactivation, which is a process that revert slowly. In addition, it is believed that ROS production decreases in the deactive MCI (Ying et al., ref. 36). Regarding this point, we think that recovery of the acute ROS and NADH hypoxic signals in Ndufs2-deficient cells by NDI1, shown in the current manuscript, clearly indicates that a hypoxia-driven conformational change of MCI is not absolutely necessary for acute O₂ sensing by glomus cells, although it may be important for the response to hypoxia in other cell types. Given these facts, which are mentioned in the manuscript (page 10, lines 11-14), we did not consider necessary to discuss the interesting paper suggested by the reviewer (which is now cited as ref. 31) in the context of our data in glomus cells.

However, we have tried to comply with the reviewer requests and have performed new experiments to test whether a Na-dependent Na/Ca exchange mechanism, proposed by Hernansanz-Agustin et al., (ref. 31) for other cell types (including tumor cell lines), is necessary for the activation of glomus cells by hypoxia. The figures shown below are presented only for reviewing purposes and are not included in the current manuscript.

i) We have tested the effect of CGP-37157 (10 μ M), a blocker of the mitochondrial Na/Ca exchanger, which at this concentration was shown by Hernansanz-Agustin et al. (ref. 31) to abolish the hypoxia-triggered Ca²⁺ signals in a breast cancer cell line. CGP-37157 produced a reversible decrease in the hypoxia-induced secretory response in both control glomus cells and in MCI-deficient cells expressing NDI1 (Fig. R2a), however a similar effect was observed in the response to high K⁺ (a way to depolarize the cells and to open voltage-gated Ca²⁺ channels independently of hypoxia) (Fig. R2b).

Figure R2. a) Glomus cell secretory activity induced by hypoxia in a KO/NDI1 carotid body slice. Note that secretory activity induced by hypoxia is reversibly inhibited by application of CGP-37157 (10 μ M). Similar effects have been observed in other 2 cells from KO/NDI1 mice and 3 cells from WT mice and

rats. b) Reversible inhibition of the secretory activity induced by high extracellular K⁺ (40 mM) in the presence of CGP-37157 (10 μM) (n= 2 cells from WT mice).

ii) The data in Fig. R2 suggested that CGP-37157 may inhibit voltage-gated Ca²⁺ channels. As expected, we have observed that CGP-37157 inhibits voltage-dependent Ca²⁺ currents in patch clamped glomus cells (Fig. R3).

Figure R3. a and b) Representative examples of whole-cell currents through calcium channels recorded in two dispersed glomus cells from WT mice in response to membrane depolarization to 10 mV from a holding potential of -70 mV. Note the inhibition of the current by the addition of CGP-37157 (10 μM). Similar effects have been observed in other 4 cells from WT mice.

These experiments indicate that, regardless of its effects on the Na⁺/Ca²⁺ exchanger, CGP-37157 also blocks membrane voltage-gated Ca²⁺ channels in glomus cells and that the effects of the drug on the secretory response to hypoxia and to high K⁺ is explained by its inhibitory effect on the Ca²⁺ channels. Please note that it has been shown by several groups (and also presented in Fig. 3e for KO/NDI1 cells) that blockade of voltage-gated Ca²⁺ channels inhibit hypoxia-induced secretion in glomus cells. Indeed, searching in the literature we have found at least two reports describing the non-selective actions of CGP-37157 that we have now directly demonstrated with patch clamp recordings (Baron and Thayer, 1997, PMID: 9537826; Ruiz et al., 2014, PMID: 24722281).

iii) We have also tested that the mitochondrial hypoxic signals in glomus cells (reversible increases in NADH and intermembrane space ROS) are recorded after complete removal of extracellular Na⁺ (Fig. R3).

Figure R4. a) Representative example of the acute hypoxic NADH signal recorded in a dispersed glomus cell bathed in a 0 Na⁺ solution (extracellular Na⁺ was completely replaced with the impermeant cation N-methyl-D-glucamine). Similar recordings have been obtained in all cells tested (n=30). b) Representative example of the acute hypoxic intermembrane space (IMS) ROS signal

recorded in a glomus cell from a CB slice bathed in a 0 Na⁺ solution (extracellular Na⁺ was completely replaced with the impermeant cation N-methyl-D-glucamine). Similar results have been obtained in all cells tested (n=8).

Taken together, these data indicate that inhibition of the Na/Ca exchanger does not seem to be necessary for the acute responsiveness to hypoxia in glomus cells. We think that the current manuscript, focused on fast acute O₂ sensing in the highly specialized glomus cells, is not the appropriate place for a detailed discussion on whether an MCI-dependent activation of the Na/Ca exchanger participates in mitochondrial responsiveness to hypoxia in other cell types (including cancer cell lines). As mentioned above, the rescue of O₂ sensing in MCI-deficient NDI1 expressing cells directly demonstrate that a conformational change in MCI is not necessary for acute O₂ sensing by glomus cells (page 10, lines 11-14).

3. Some parts of the model described (Fig. 4 i,j) are not easily understood or well supported.
a. It is strange that in WT cells MCI produce ROS directly to the IMS. Most models of superoxide production by MCI locate it directed to the matrix, while IMS superoxide production is driven mainly by the Qo site of complex III.

Authors: The production of ROS directed to the mitochondrial intermembrane space (IMS) in MCI was discussed in detail in our previous publications (Fernandez-Aguera et al., ref. 7; Arias-Mayenco et al., ref. 8). We proposed that during hypoxia accumulation of QH₂ leads to production of superoxide at the MCI Q site that is repelled by the electrostatic field in the inner membrane (negative the matrix respecting the IMS) to the IMS where it is converted to H₂O₂. In normal glomus cells, matrix ROS always decreases during exposure to hypoxia. We also think that a component of the hypoxic IMS ROS signal comes from MCIII. This is the way ROS production is indicated in our published models as they are represented in Fig. 4a and b.

b. In the NDI1 cells, why would the matrix dehydrogenases be less active and produce less ROS? There is matrix NADH accumulation and ROS increase, so they could be more active and producing more superoxide.

Authors: In all the studies that we have done on ROS production by glomus cells it was observed that hypoxia produces a highly reversible decrease in ROS levels at the mitochondrial matrix. Our interpretation (in agreement with other investigators, see Arias-Mayenco et al., 2018, ref. 8), is that although the activity of matrix dehydrogenases is probably unaltered during hypoxia, the decrease in O₂ availability leads to a non-selective reduction in ROS production. A new finding now is that in NDI1-expressing cells, we systematically record a robust increase in matrix ROS during hypoxia. We propose that this signal is caused by electrons escaping from NDI1, due to slow down/reversion of enzyme activity secondary to the increase in QH₂ levels.

c. In the NDI cells, why are IMS ROS assumed to come from the matrix? They could be produced directly to the IMS by complex III, for example, even in less extent than in the WT cells.

Authors: It is believed that ROS (probably in the form of H₂O₂) produced in the matrix can move rapidly to the intermembrane space. For example, in glomus cells rotenone can produce a ROS signal at the matrix which can be recorded at the intermembrane space (see Arias-Mayenco et al., 2018, ref. 8). This is the reason why we have suggested that the increase in intermembrane ROS induced by hypoxia in NDI1-expressing cells could come from the matrix. However, we think that, as pointed out by the reviewer, part of the intermembrane ROS signal is generated at MCIII. This is already indicated in the diagrams in Fig. 4 (a,b). To clarify this point, we have added a reference to these diagrams in the text (page 8, line 6 from the bottom).

4. The paper assumes that the presence of NDI1 overcomes NADH:CoQ activity of MCI. It is an interesting suggestion, but in that case the proton gradient and OXPHOS function could also be affected. It could be worthy trying to confirm this point by measuring the proton gradient or the mitochondrial membrane potential, or any other way to assess specifically the function of MCI.

Authors: We have done experiments to evaluate the contribution of OXPHOS to ATP production in the various genetically modified glomus cells studied using a fluorescent probe. We show that NDI1 restores production of ATP by OXPHOS (Fig. 2e,f). The fact that NDI1 overcomes NADH:CoA activity of MCI is strongly supported by the lack of effect of rotenone in WT/NDI1 glomus cells (Supplementary Fig. 8b,g,h and Supplementary Fig. 9d-f). We think, however, that a detailed description of the interaction of NDI1 with MCI and other mitochondrial components, which is a topic being currently investigated in other laboratories, is outside the scope of the current manuscript.

5. Another assumption of the paper is that NDI1 is reversed or slowed down during hypoxia. Could this be assessed experimentally? Maybe using NDI1 inhibitors, or forcing reversal in normoxia providing an excess of succinate...

Authors: A slow down/reversal of the NADH dehydrogenase activity of MCI (WT glomus cells) or NDI1 (KO/NDI1 glomus cells) is concluded given the fast and reversible increase in NADH recorded during exposure to hypoxia. Experiments suggested by the reviewer have been done before in WT cells (Fig. 6 in Arias-Mayenco et al., ref. 8). We have checked the effect of flavone, which inhibits NDI1 (Boo Seo et al. PMID:10982813), but we discarded these data because the drug seems to have non-selective effect on mitochondrial parameters (NADH signal) in WT glomus cells.

6. Also interesting is the fact that NDI1 expression affects CO₂ sensing in both KO/NDI1 and WT/NDI1 cells. In p. 8 the authors suggest an explanation based on the Na⁺/H⁺ antiporter activity of deactive MCI. Would this activity of MCI be maintained in the KO/NDI1 cells?

Authors: It has been shown that the transmembrane arm (TA) of deactive MCI is a Na/H antiporter (Robert and Hirsch, 2012; ref. 37). We have also shown that although the full MCI is not assembled in Ndufs2-deficient cells, there is a molecular complex in these cells with a molecular weight compatible with MCI TA which is positive for antibodies against ND1 (a transmembrane protein part of the MCI TA) (see Fig. 5a in Fernandez-Aguera et al., ref. 7). Therefore, it is plausible that in KO/NDI1 cells the assembled MCI TA (without any NADH dehydrogenase activity) confers upon their mitochondria a Na/H antiporter activity similar to the one described for the deactive MCI.

Minor points:

p.3. The Ndufs2 KO mice are described as (Ndufs2 flox^{-/-};Cre), are they homozygous or heterozygous for Ndufs2 flox and Cre-dependent deletion?

Authors: To facilitate an efficient deletion of the floxed Ndufs2 allele we used flox^{-/-} mice with only a copy (heterozygous) of Cre. This is now specified in the Methods section (page12, first 6 lines of the Methods section). We have already tested that the flox^{-/-} mice have normal responses to hypoxia (Fernandez-Aguera et al., ref. 7) as well as other general physiological functions (McElroy et al., 2022; PMID: 35338200).

p.4 and ED Fig. 2c & 9g. Please describe what type of activity of MCI is measured, as it is supposed to be different from the NADH:CoQ oxidoreductase activity that is also exerted by NDI1.

Authors: We used a commercial dipstick assay described in our previous publications (Fernandez-Aguera et al., ref. 7; Arias Mayenco et al., ref. 8). This method measures NADH dehydrogenase activity after immunoprecipitation of MCI on the dipstick. To clarify this point a sentence has been added to the Method section (page 18, MCI activity section)

p. 4 explains Fig. 1f as “hypoxia-induced lactatemia”, but the figure is described as basal blood lactate.

Authors: It was a mistake, which has been corrected.

p. 6 and Fig. 4 c,g. It would be good to say in the text that the IMS ROS signal in KO-NDI1 is lower than in WT cells (the scale in Fig. 4c is different than in Fig. 4a).

Authors: There was not statistically significant difference in the IMS ROS signal between WT and KO/NDI1 cells (Fig. 5g in the revised manuscript).

Fig. 4i is not explained in the legend.

Authors: This is now corrected.

ED Fig. 8c. It would be better to show the median, as in Fig. 3j. Also ED Fig 8h,j would be better in the same scale and graph type as in Fig. 4g,h.

Authors: The data in ED Fig. 8b,c (Extended Data Fig. 9b,c in the revised manuscript) follow a normal distribution and for that reason the two plots are presented as mean plus scatter diagram. Following the indication of the reviewer, we have changed the representation of data in ED Fig. 8j to a box plot diagram. Now, the two plots (Extended Data 9h,j in the revised manuscript) have the same type of representation and can be more easily compared

Reviewer #4 (Remarks to the Author):

The study by Jimenez-Gomez and colleagues extends their previous work examining the function of MCI and its NADH dehydrogenase activity on acute O₂ sensing of the carotid body glomus cells. They show, as they have previously, that knockout mice lacking MCI specifically in TH⁺ cells have a nearly absent hypoxic response which is also associated with abolished dopaminergic neurosecretion (by CB glomus cells in vitro), NADH and ROS accumulation. All of these effects are reversed in transgenic mice that express a yeast NADH dehydrogenase (ND1) driven by the TH promoter. While the data are sound, I have a number of serious concerns that should be addressed.

Authors: We thank the reviewer for the positive comments on our work. We also thank him/her for the time spent in a detailed evaluation of our manuscript and the insightful suggestions, which have helped us to improve the quality of the work. We have performed new experiments and added new data and explanations to the revised manuscript to address all the questions raised by the reviewer. With the new data the manuscript has been expanded and adapted to the format of Nature Communications. We have also added a new author (Dr. Patricia González-Rodríguez), who collaborated in the generation of the conditional *Ndufs2* KO mice. More recently, she has also contributed to the project in the study of ATP levels in single glomus cells using two-photon microscopy and a genetically encoded fluorescent probe.

1. The authors show absent hypoxic responses in KO mice, with restoration in ND1 Tg mice. ND1 is expressed in all TH+ cells, so how can the authors be sure that the restoration of the HVR is specific to the glomus cells?

Indeed, catecholaminergic neurons are distributed throughout the neural circuitry involved in the HVR, including the NTS, pons, VLM etc. Did the authors consider an effect at these other sites?

Authors: We understand and fully agree with the comment of the reviewer. However, it must be kept in mind that the primary objective of our research is to investigate the mechanisms of acute O₂ sensing by CB glomus cells and in this structure only glomus cells are TH positive. We also attempt to correlated changes in glomus cell function with the HVR and for this reason in all our experiments on genetically modified mouse models we do control measurements at the systemic and single cell levels.

In the case of the HVR, we have recorded in parallel the hypercapnic ventilatory response, not only to make sure that the central respiratory circuits are working well, but also to show that activation by hypercapnia of peripheral (mainly the CB) and central chemoreceptors is not altered. We have added a short sentence to the text stressing the importance of these control experiments (page 4, last 4 lines). We have shown in previous publications that ablation of genes affecting the mitochondrial electron transport chain using Cre recombinase driven by the TH promoter does not seem to produce significant alterations in respiratory control, other than inhibition of the O₂ sensing mechanism in peripheral chemoreceptors (e.g., Fernandez-Aguera et al., ref.7; Cabello-Rivera et al. ref. 17).

At the cellular level, our previous work and current data clearly indicate that genetic disruption of MCI produces a selective abolition of responsiveness to hypoxia maintaining the responses to hypercapnia, hypoglycemia or lactate unaltered (Fernandez-Aguera et al. ref. 7; Torres-Torrelo et al., ref. 20). In the current report we show that the response to hypoxia is rescued by expression of NDI1 in glomus cells.

The link between CB sensitivity to hypoxia and the HVR is very well documented in the literature. For example, CB atrophy or denervation (tested in humans and in rodents) practically abolishes the HVR (e.g., Macias et al., PMID: 29671738; Timmers et al. PMID: 14528027; Limberg et al. PMID: 25870188; Niewinski et al., PMID: 33595103). Therefore, it is very unlikely that expression of NDI1 in TH-positive cells other than glomus cells is sufficient for the rescue of the HVR in KO/NDI1 mice.

An additional consideration is that glomus cells are multimodal sensory receptors and although cells without the O₂ sensing system (due to MCI disruption) are insensitive to hypoxia, they are able to sense other stimuli (e.g., hypercapnia), which by activating the same output line produce hyperventilation. This indicates that the various steps in the output line (glomus cell---afferent fibers---brainstem neurons---hyperventilation) are maintained functional.

Although we understand the reviewer caveats, previous knowledge and data available strongly support the view that the inhibition the HVR in our KO (MCI-deficient) mice model is mainly due to the loss of hypoxia sensitivity in glomus cells and that the expression of NDI1 in these cells is the main responsible for recovery of the HVR is KO/NDI1 mice.

2. Related to concern 1 above. It would be more convincing to show effects of KO and ND1 rescue at the level of the CSN afferents.

Authors: As it was indicated above, the primary objective of our research is to investigate the mechanisms of acute O₂ sensing in CB glomus cells. To this end we have developed over the years preparations (dispersed cells and CB slices) that allowed us to do single-cell physiology experiments (amperometry and microfluorimetry, among others). These preparations also provide direct access to the extracellular medium and therefore the external solution composition can be precisely controlled. The recording of afferent sensory

fibers is an interesting preparation but does not make feasible to perform the type of analytical experiments at the single-cell level mentioned above. In addition, the *in vitro* whole CB-afferent fiber preparation does not allow precise control of the composition of the extracellular medium.

As it stands, the only data shown related to glomus cell response is dopamine secretion. However, these data are from isolated glomus cells that are exposed to PO₂ of 10-15mmHg (i.e. not physiological).

Authors: In addition to dopamine secretion, we also show in this manuscript microfluorimetric measurements of NADH levels and real-time ROS production at the mitochondrial matrix and intermembrane space. These are valuable data that robustly support our main conclusions. To address the reviewer comment, we have done a new set of experiments on dispersed Fura-2-loaded glomus cells to show the suppression of the hypoxic cytosolic calcium signal in KO cells and its recovery in NDI1 expressing cells (Supplementary Fig. 5; page 6 last line and page 7, first lines).

It is true that the we normally use a strong hypoxic stimulus (PO₂ ~10-15 mmHg), however it must be kept in mind that these are PO₂ values at the solution in contact with the dispersed cells or cells at the slice surface, which in physiological conditions is only reached with much higher arterial PO₂ values. For example, the HVR in rodents is normally tested with 5-10% O₂ (35-70 mmHg) in inspired air (10% in the current manuscript). These values in inspired air result in arterial PO₂ levels in arterial blood in the range of 25-50 mmHg, respectively (see d'Anglemon et al., PMID: 27774479). It is most likely that with these arterial O₂ tensions, the O₂ tension in the vicinity of glomus cells is even lower than the values used in our *in vitro* experiments.

We have now done new experiments to show that recovery of glomus cell hypoxic secretory response by NDI1 expression is observed over a broad range of O₂ tensions (Supplementary Fig. 4; page 6 last four lines).

Moreover, it is apparent in their own data that neurosecretion *in vitro* may correlate poorly with the strength of the conscious animal (compare Suppl. Figs 6 (whole animal response of WT/ND1 mice) with Suppl. 7 (secretory response from WT/ND1 glomus cells)).

Authors: In general, we observe a good correlation between CB activation *in vitro* and the HVR *in vivo* although as it was indicated by the reviewer in addition to CB input there are other process that may influence the HVR in conscious animals, which are not considered in this study. We have systematically observed in cells expressing NDI1 a slight increase in the secretory response to hypoxia in comparison with WT cells. However, these differences are not appreciated *in vivo*. In our experimental conditions (no CO₂ added to the inspired gas used to test HVR) hyperventilation induced by hypoxia produces hypocapnia, which blunts and attenuates the HVR. These changes in CO₂ levels do no occur *in vitro* and for this reason the glomus cell secretory response to hypoxia is not blunted.

3. As the KO is across all TH+ neurons, are the authors sure that the decreased HVR may be related to altered metabolic drive? I ask because Fig. 1d and Suppl. Fig 10 suggest reduced respiratory frequency at baseline, suggesting reduced drive. Are their any differences in body temperature between the WT, KO and KO+ND1 mice?

Authors: Following the reviewer suggestion we have measured body temperature and found it to be slightly decreased in KO and KO/ND1mice in comparison with WT mice (Supplementary Fig. 2d). A comment on the potential effect of this and other parameters on the basal respiratory frequency has been added to the text (page 4, lines 4-9 from the bottom).

4. Fig. 1b suggests that the KO/ND1 mice have a very transient HVR (over about 3 time bins

or ~20 seconds). Was this analyzed, and how come it is not reflected in the data shown in Suppl. Fig. 3 (peak response).

Authors: As indicated in Methods peak respiratory frequency was measured by averaging the values of 20 points (40 s) around the peak of the hypoxic response. These values were practically similar in WT and KO/NDI1 mice (Supplementary Fig. 1a). Average respiratory frequency was in most cases calculated by averaging 40 points (80 seconds) after reaching 10% O₂ tension in the chamber and these data were those presented in Supplementary Fig. 1b (now marked as Hx1 in the revised figure). Whereas peak respiratory frequency was slightly higher in KO/NDI1 mice (247±5 breaths/min) than in WT mice (241±7 breaths/min) mice, the average respiratory frequency was smaller in KO/NDI1 (228±5 breaths/min) than in WT (236±8 breaths/min) mice. We have now extended the analysis and calculated average breathing frequency covering practically the entire exposure to hypoxia (averaging the values during 300 seconds after reaching 10% O₂ tension in the chamber before returning to normoxia). These new data are presented as Hx2 in Supplementary Fig. 1b.

Minor:

1. It would be nice to see actual evidence of ND1 expression, in addition to GFP immunofluorescence.

Authors: Following the reviewer suggestion we have confirmed the increase in mRNA NDI1 expression by PCR (see Fig. 2b).

2. Some of the figures have data for "Sdhd", but I could not find a definition for this molecule.

Authors: Corrected. Sdhd is now defined when it is first mentioned (Supplementary Fig. 3 legend).

REVIEWERS' COMMENTS

Reviewer #1 (Remarks to the Author):

The authors have provided adequate responses to my concerns, I support the publication of the paper. .

Reviewer #3 (Remarks to the Author):

I appreciate the effort of the authors in improving the manuscript in response to my comments and those of the other reviewers. Even though some of my suggestions have not been fully addressed, the main message of the manuscript is clearly demonstrated and provides a very interesting contribution in the field. As I already stated, the paper is very interesting for being published in Nature Communications, which will enhance the discussion about the topic, including some aspects of the model proposed that merit further research.

I still have some minor suggestions that could be taken into account, but that not preclude publication of the paper. I follow the same numbers of my previous comments.

3. The model proposed by the authors is coherent with their previous proposals and provide a good explanation of the results obtained. I appreciate very much their explanations to my questions, and I realize that there are several parts of the model which are still not fully confirmed, even though there are many groups working un the field. I just point out that the schemes in Fig. 4a-b and Fig. 5i-j show different locations for ROS production (if Figs 4a-b are interpreted in terms of IMM-separated localizations), and I wonder if the authors want to modify the schemes so they are more coherent among them, or they prefer to show these differences in a slight detail of the mechanistic models they provide.

Minor points, 1st (p.3). I understand that they have used heterozygous animals for Cre, and I suppose that they are homozygous for the flox construct of Ndufs2, so when the animals are positive for Cre, they will recombine and knock out the Ndufs2 gene in both copies of the genome. I just wonder if the notation they use (Ndufs2 flox^{-/-};Cre) could be understood as the animals being heterozygous for the flox construction of Ndufs2.

Minor points, 2nd (p.4 and ED Fig 2c & 9g). I appreciate that the authors have better described Ci activity measurement in page 18, so it can be better understood the specificity of this measurement for CI and not for ND1. However, they state that “the resulting H⁺ reduced NBT” which seems to be a chemical error (how could a H⁺ with no electrons behave as a reducing agent providing electrons?). I suggest to remove “the resulting H⁺” for a better meaning of the sentence.

Reviewer #4 (Remarks to the Author):

The authors have done a nice job responding to all concerns and revising the manuscript accordingly. Congratulations on a well done study.

RESPONSE TO THE REVIEWER'S COMMENTS

Jiménez-Gómez et al.

Authors: We thank the reviewers for the time spent in the evaluation of our paper and for their insightful comments and suggestions.

Reviewer #1 (Remarks to the Author):

The authors have provided adequate responses to my concerns, I support the publication of the paper.

Authors: No further responses are required.

Reviewer #3 (Remarks to the Author):

I appreciate the effort of the authors in improving the manuscript in response to my comments and those of the other reviewers. Even though some of my suggestions have not been fully addressed, the main message of the manuscript is clearly demonstrated and provides a very interesting contribution in the field. As I already stated, the paper is very interesting for being published in Nature Communications, which will enhance the discussion about the topic, including some aspects of the model proposed that merit further research.

Authors: We thank the reviewer for the positive comments on our paper and for the new suggestions, which have been taken into consideration in the revised version of the manuscript.

I still have some minor suggestions that could be taken into account, but that not preclude publication of the paper. I follow the same numbers of my previous comments.

3. The model proposed by the authors is coherent with their previous proposals and provide a good explanation of the results obtained. I appreciate very much their explanations to my questions, and I realize that there are several parts of the model which are still not fully confirmed, even though there are many groups working un the field. I just point out that the schemes in Fig. 4a-b and Fig. 5i-j show different locations for ROS production (if Figs 4a-b are interpreted in terms of IMM-separated localizations), and I wonder if the authors want to modify the schemes so they are more coherent among them, or they prefer to show these differences in a slight detail of the mechanistic models they provide.

Authors: We very much appreciate this comment of the reviewer. We had drawn the inner mitochondrial membrane (IMM) in the wrong place. We have modified Fig. 4a,b to place ROS production in the right location.

Minor points, 1st (p.3). I understand that they have used heterozygous animals for Cre, and I suppose that they are homozygous for the flox construct of Ndufs2, so when the animals are positive for Cre, they will recombine and knock out the Ndufs2 gene in both copies of the genome. I just wonder if the notation they use (Ndufs2 flox^{-/-};Cre) could be understood as the animals being heterozygous for the flox construction of Ndufs2.

Authors: We used heterozygous mice with one floxed allele and one minus (-) allele (Ndufs2

flox^{-/-};Cre). After Cre-mediated recombination the mice were ^{-/-}. We think this is explained in the text and that no further change is needed.

Minor points, 2nd (p.4 and ED Fig 2c & 9g). I appreciate that the authors have better described Ci activity measurement in page 18, so it can be better understood the specificity of this measurement for Ci and not for ND1. However, they state that “the resulting H⁺ reduced NBT” which seems to be a chemical error (how could a H⁺ with no electrons behave as a reducing agent providing electrons?). I suggest to remove “the resulting H⁺” for a better meaning of the sentence.

Authors: In the sentence “...oxidized NADH and the resulting H⁺ reduced NBT to form a...” we have removed “the resulting H⁺” as suggested by the reviewer (page 18, line 2 from the bottom).

Reviewer #4 (Remarks to the Author):

The authors have done a nice job responding to all concerns and revising the manuscript accordingly. Congratulations on a well done study.

Authors: No further responses are required. We thank the reviewer for the positive comments on our paper.